# Understanding the Emergence of Multimodal Representation Alignment

**Megan Tjandrasuwita** [1]   **Chanakya Ekbote** [1]   **Liu Ziyin** [1 2]   **Paul Pu Liang** [1]

## Abstract

Multimodal representation learning is fundamentally about transforming incomparable modalities into comparable representations. While prior research primarily focused on *explicitly* aligning these representations through targeted learning objectives and model architectures, a recent line of work has found that independently trained unimodal models of increasing scale and performance can become *implicitly* aligned with each other. These findings raise fundamental questions regarding the emergence of aligned representations in multimodal learning. Specifically: (1) when and why does alignment emerge implicitly? and (2) is alignment a reliable indicator of performance? Through a comprehensive empirical investigation, we demonstrate that both the emergence of alignment and its relationship with task performance depend on several critical data characteristics. These include, but are not necessarily limited to, the degree of similarity between the modalities and the balance between redundant and unique information they provide for the task. Our findings suggest that alignment may not be universally beneficial; rather, its impact on performance varies depending on the dataset and task. These insights can help practitioners determine whether increasing alignment between modalities is advantageous or, in some cases, detrimental to achieving optimal performance. Code is released at: https://github.com/MeganTj/multimodal_alignment.

## 1. Introduction

Multimodal AI represents a cutting-edge paradigm in machine learning that enables integrating and learning from many heterogeneous and interacting data modalities. These

[1]Massachusetts Institute of Technology, USA [2]NTT Research, USA. Correspondence to: Megan Tjandrasuwita <megantj@mit.edu>.

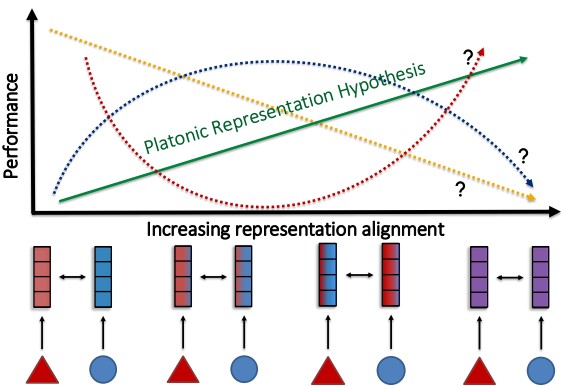

Figure 1: **Emergence of multimodal alignment?** Triangles and circles correspond to different modalities. While the Platonic Representation Hypothesis (Huh et al., 2024) argues that better cross-modal alignment predicts better performance, our findings demonstrate that the relation between alignment and performance is more nuanced and depends on several dataset characteristics including the degree of heterogeneity and interactions between modalities.

AI systems are revolutionizing predictive analytics across many applications, including in multimedia (Alayrac et al., 2022; Sun et al., 2019; Ramesh et al., 2021; Singer et al., 2022), healthcare (Cai et al., 2019; Muhammad et al., 2021), and physical sensing (Kirchner et al., 2019; Lee et al., 2019; Xiao et al., 2020). A large body of research in designing and training multimodal models has focused on *aligning* the representations from different modalities such that they are comparable in some semantic representation space (Baltrušaitis et al., 2018; Liang et al., 2024). Conventional wisdom posits that aligned representations are a crucial precursor to multimodal fusion and representation learning (Li et al., 2021). As a result, many learning methods, such as contrastive learning and its variants (Frome et al., 2013; Jia et al., 2021; Radford et al., 2021a), and model architectures (Bertinetto et al., 2016; Lenc & Vedaldi, 2019; Bansal et al., 2021; Csiszárik et al., 2021) have been proposed to explicitly align incomparable modalities into comparable representation spaces for further processing.

However, recent work on the "Platonic Representation Hypothesis" showed that, surprisingly, alignment could even emerge across independently pre-trained vision and language models without explicitly aligning them to-

gether (Huh et al., 2024). Crucially, alignment increases with model size and performance, and it has been hypothesized that unimodal models will become increasingly aligned. These findings raise fundamental questions regarding the emergence of aligned representations and their implications on multimodal learning: (1) when and why does alignment emerge implicitly, and (2) is alignment a reliable indicator of performance? We illustrate these open questions in Figure 1.

In this paper, we study these questions comprehensively across two principal dimensions that taxonomize multimodal data: *interactions* and *heterogeneity* (Baltrušaitis et al., 2018; Liang et al., 2024; Tian et al., 2020), visualized in Figure 2. Interactions measure the information shared between two modalities for a task, from more redundant (e.g., images and corresponding captions) to more unique (e.g., sensor placement). We expect alignment to emerge more easily between redundant modalities. Heterogeneity measures the degree of similarity across two modalities independent of the task, from more similar (e.g., two languages) to more different (e.g., text and video). We expect alignment to emerge more easily between similar modalities.

Through extensive experiments on controlled and real-world datasets with varying degrees of interactions and heterogeneity, we discover several key insights. First, the maximum alignment achievable depends on the degree of heterogeneity and uniqueness in the modalities, which inherently limits alignment. Second, while alignment correlates with performance in datasets with high redundancy, this relationship breaks down when uniqueness dominates redundancy. These findings highlight that performance often does not directly correspond to alignment, and the connection between them is a nuanced property of the data that varies across modalities and tasks. Therefore, our work provides important considerations for practitioners designing and training multimodal models, emphasizing that scale alone does not guarantee modality alignment and that careful assessment is necessary to determine when alignment is beneficial.

## 2. Representation Alignment

In this section, we review the concept of representation alignment through prior work on measuring alignment, methods to explicitly align representations, and observations regarding the emergence of alignment.

**Measuring Alignment.** Measuring alignment between neural network representations is a widely used approach in the research community to analyze and improve training dynamics (Huh et al., 2024; Klabunde et al., 2024; Kornblith et al., 2019). A prominent class of alignment metrics is based on canonical correlation analysis (CCA), a statistical technique for comparing two subspaces (Thompson, 2005;

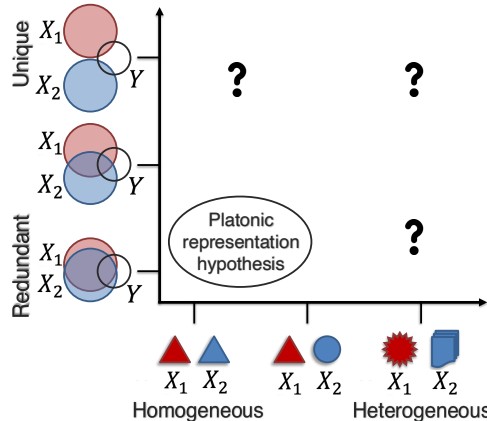

Figure 2: **Two principal dimensions of multimodal data.** This study empirically evaluates data across two key dimensions: heterogeneity and interactions. Heterogeneity, represented on the x-axis, reflects the similarity between two data modalities, $X_1$ and $X_2$, regardless of the task. Interactions, on the y-axis, indicate the balance between redundant and unique information across modalities that is relevant to task $Y$. We expect the Platonic Representation Hypothesis to hold in cases of redundancy and similar modalities, but when and why alignment emerges implicitly, and whether alignment is a reliable indicator of performance, remain open questions.

Golub & Zha, 1995), along with its nonlinear extensions using kernels (Lai & Fyfe, 2000; Raghu et al., 2017) and neural networks (Andrew et al., 2013; Wang et al., 2015; Morcos et al., 2018). Among these methods, Kornblith et al. (2019) highlight the advantages of Centered Kernel Alignment (CKA), particularly its invariance to orthogonal transformations and isotropic scaling. Given mean-centered feature sets of $n$ samples, $Z_1, Z_2 \in \mathbb{R}^{n \times d}$, from two modalities $X_1$ and $X_2$, the CKA metric with a linear kernel is:

$$\mathsf{CKA}(Z_1, Z_2) = \frac{\mathsf{ALIGN}(Z_1, Z_2)}{\sqrt{\mathsf{ALIGN}(Z_1, Z_1) \cdot \mathsf{ALIGN}(Z_2, Z_2)}}$$

where $\mathsf{ALIGN}(Z_1, Z_2)$ denotes $\mathsf{HSIC}(Z_1 Z_1^\mathsf{T}, Z_2 Z_2^\mathsf{T})$ with HSIC denoting an empirical estimator of the Hilbert-Schmidt Independence Criterion (Gretton et al., 2005). Intuitively, CKA quantifies alignment by comparing the covariance structures of two feature sets, capturing whether their representations encode similar relationships. This property ensures that even if $Z_1$ and $Z_2$ undergo arbitrary rotations (e.g., due to different initialization schemes), the CKA metric remains consistent, making it a robust choice for assessing representation alignment. For large vision-language models, given the high dimensionality and large embedding sizes of learned representations, we employ a

computationally efficient variation of CKA — the mutual k-nearest neighbors (mutual KNN) method (Huh et al., 2024). Instead of directly comparing full covariance structures, this approach measures similarity by analyzing the overlap between the k-nearest neighbor sets of embeddings, improving scalability. Details are provided in Appendix B.

**Explicit Alignment.** In addition to research on measuring alignment in neural networks, another line of work focuses on explicitly aligning representations, a widely used technique for handling heterogeneous modalities (Liang et al., 2024). A popular approach in this domain is multimodal contrastive learning, where representations of the same concept across different modalities (i.e., positive pairs) are brought closer together, while representations of different concepts (i.e., negative pairs) are pushed apart (Frome et al., 2013; Jia et al., 2021; Radford et al., 2021a). The coordination distance in contrastive learning is typically measured using cosine distance (Mekhaldi, 2007) or max-margin losses (Hu et al., 2019). Theoretical results demonstrate that contrastive learning effectively captures redundant information shared between modalities (Tian et al., 2020; Tosh et al., 2021). More recent extensions have been proposed to also capture unique and synergistic information, further refining multimodal representation learning (Dufumier et al., 2024; Liang et al., 2023b).

**Emergence of Implicit Alignment.** In contrast to explicit alignment methods, recent findings suggest that alignment can emerge implicitly, even when neural networks differ in training objectives, datasets, and architectures (Li et al., 2015; Raghu et al., 2017; Lenc & Vedaldi, 2019; Barannikov et al., 2022; Bonheme & Grzes, 2022). Notably, this similarity becomes more pronounced in larger and wider networks (Raghu et al., 2017; Morcos et al., 2018; Kornblith et al., 2019). Building on the observation that latent spaces are inherently comparable, a line of research explores composing components of different models with minimal or no additional training. Lenc & Vedaldi (2019) demonstrate that latent spaces can be stitched together using trainable stitching layers, while subsequent studies (Bansal et al., 2021; Csiszárik et al., 2021) show that better-performing models tend to learn more similar representations when stitched. More recently, the Platonic Representation Hypothesis (Huh et al., 2024) suggests that as vision and language models scale in capacity and performance, independently trained models exhibit increasing alignment. This finding implies that models are converging toward modality-agnostic representations, reinforcing the idea that alignment may emerge naturally as a byproduct of model scaling. However, if alignment continues to emerge, there would be no need for any of the explicit alignment methods described above. That explicit alignment has consistently been helpful implies either emergent alignment is not sufficient or emergent alignment

does not always lead to improved performance. This discrepancy between the possibility of emergent alignment and the need for explicit alignment methods calls for a systematic exploration of the role of alignment and its downstream relationship to performance in multimodal learning.

## 3. Research Questions and Experimental Setup

The recent line of work on the emergence of alignment across independently pre-trained unimodal models raises fundamental questions regarding the emergence of aligned representations and their implications on multimodal learning. Our research seeks to understand (1) when and why alignment emerges implicitly, and (2) whether alignment is a reliable indicator of performance. To reliably and comprehensively study these questions across all types of multimodal data, we use two principle dimensions to taxonomize multimodal data: *interactions* and *heterogeneity* (Baltrušaitis et al., 2018; Liang et al., 2024; Tian et al., 2020). Interactions measure the task-relevant information shared between two modalities. We expect alignment to emerge more easily between modalities where information content is redundant. Heterogeneity measures the degree of similarity across two data modalities independent of the task, from more similar (e.g., two languages) to more different (e.g., text and video). We expect alignment to emerge more easily between similar modalities. Our experiments aim to study the emergence of alignment and its relationship to downstream task performance by systematically varying the interactions and heterogeneity in multimodal data. To summarize, our fundamental guiding questions are:

1. Does alignment emerge when uniqueness and heterogeneity increase?
2. Does higher alignment always predict better performance when uniqueness is present?
3. How can we characterize datasets through the correlation between performance and alignment?

Based on these questions, we define our problem setting.

### 3.1. Problem Setting

We focus on a simplified setting with two modalities and an associated label, the generalization is straightforward. Concretely, we consider a scenario where we sample multimodal data and labels $x_1, x_2, y \sim \mathbb{P}(X_1, X_2, Y)$ from a data distribution $\mathbb{P}(X_1, X_2, Y)$. $X_i$ represents the random variable for the $i$-th modality and $Y$ for the task. Based on the relationships between $X_1$, $X_2$, and $Y$, these modalities can exhibit different degrees of interactions and heterogeneity.

**Interactions** measure the information shared between two modalities for a task, from more redundant to more unique. Redundancy $R$ represents the shared information between the two modalities and the task ($Y$), such as between images

and captions that describe the image (Radford et al., 2021a). Uniqueness in modality 1 ($U_1$) quantifies the amount of information present in the first modality absent in the second but critical for the downstream task (and likewise for $U_2$). For example, feature selection is often optimized to provide new unique information and minimize redundancy to previous ones (Peng et al., 2005).

To investigate how alignment and performance change with respect to different interactions, we need synthetic controllable datasets and real-world multimodal benchmarks with different interactions. For constructing synthetic data, we assume that the task-relevant information (for a particular label $y$) can be decomposed into $x_r, x_{u_1}, x_{u_2}$, where $x_r$ denotes the common or redundant information, $x_{u_1}$ represents information unique to the first modality, and $x_{u_2}$ captures information unique to the second modality.

We construct the input data as $x_1 = [x_r, x_{u_1}]$ and $x_2 = [x_r, x_{u_2}]$. An overview of the data generation process is shown in Figure 3. By selecting specific features to compute the label, we control the levels of redundancy and uniqueness. Specifically, $Y$ is a nonlinear function of a subset of features, $\mathcal{S} \subseteq [x_r, x_{u_1}, x_{u_2}]$. This enables us to control $R$ as the number of features in $\mathcal{S}$ that come from $x_r$, and $U_i$ as the number of features that come from $x_{u_i}$ for $i \in \{1, 2\}$. We denote the total uniqueness $U = |\mathcal{S}| - R$. By keeping $|\mathcal{S}|$ fixed while varying $U$, we generate datasets with different proportions of redundant versus non-redundant information.

**Heterogeneity**. Different modalities often exhibit distinct structures, qualities, and representations (Liang et al., 2024). For example, when one modality is a time series and another is a static image, differences in their vocabulary tokens, and different noise or distribution shifts in each modality. We aim to investigate how alignment and performance change with different degrees of heterogeneity, from more similar (e.g., two languages) to more different (e.g., text and video).

To generate synthetic datasets with varying heterogeneity, we start with the case where both modalities are redundant, meaning $Y$ (the labels) is a nonlinear function of $x_r$. Specifically, let $x_1 = x_r$ and $x_2 = \phi(x_r)$, where $\phi(\cdot)$ is a nonlinear function, as shown in Figure 3. In this setting, heterogeneity is defined as the number of nonlinear transformations involved in $\phi(\cdot)$, and we assume that nonlinear transformations are bijections, ensuring that the information content of the heterogeneous modality remains unchanged. Concretely, if $\phi(\cdot)$ is modeled as a multilayer perceptron (MLP), the number of layers $D_\phi$ quantifies the level of heterogeneity between the two modalities. We extend this definition to cases where the modalities contain unique information. Let $X_1 = [x_r, x_{u_1}]$, and a modality that is heterogeneous with respect to $X_1$ is defined as $X_2 = \phi([x_r, x_{u_2}])$.

**Experimental setup for synthetic datasets.** We evaluate

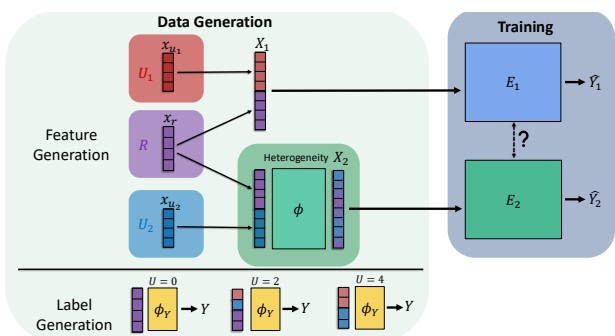

Figure 3: **Synthetic data generation and training.** We generate synthetic data with varying levels of uniqueness and heterogeneity. The building blocks are the redundant and unique components $[x_r, x_{u_1}, x_{u_2}]$, where $[x_r, x_{u_1}]$ are used in creating $X_1$ and $[x_r, x_{u_2}]$ are for $X_2$. The level of uniqueness is determined by the number of features from $x_r$ that are used to compute the labels $Y$, given that the total number of features used for label computation is held constant. $X_2$ is transformed into a heterogeneous modality using a transformation network $\phi$. In our experiments, we compute alignment between unimodal encoders $E_1$, $E_2$ trained on $X_1$, $X_2$ respectively.

how uniqueness, redundancy, and heterogeneity influence the emergence of alignment by training encoders independently on each modality and measuring the alignment between their learned representations. Specifically, we train a single-layer encoder on the first modality, denoted as $E_1$. We experiment with higher depths of $E_1$ in Appendix D.2 and find that the results are not significantly changed. For the second modality, which is transformed by the nonlinear function $\phi(\cdot)$ with varying depths ($D_\phi$), we train a series of encoders denoted as $E_{2,D_{Enc}}$, where $D_{Enc}$ represents the depth of the encoder trained on the second modality and varies as $D_{Enc} \in \{1, \ldots, 10\}$.

**Experimental setup for real benchmarks.** In addition to experiments on synthetic data, we conduct analogous experiments on vision-language models. We use the same dataset and models as Huh et al. (2024), which evaluates alignment on the Wikipedia caption dataset (Srinivasan et al., 2021) with naturally co-occurring text and images. This dataset is inherently heterogeneous (text and images are different) with high redundancy due to overlapping semantic information. To vary the amount of unique information, we leverage GPT-4 to synthesize text captions with unique information that is not present in the images. For each (image, text) pair in the original dataset, we prompt GPT-4 to produce 10 captions with increasing levels of uniqueness: 10%, 20%, … 100%, such that the final caption contains only information that is unique to the text. As uniqueness is already introduced in the text, we keep the original images in

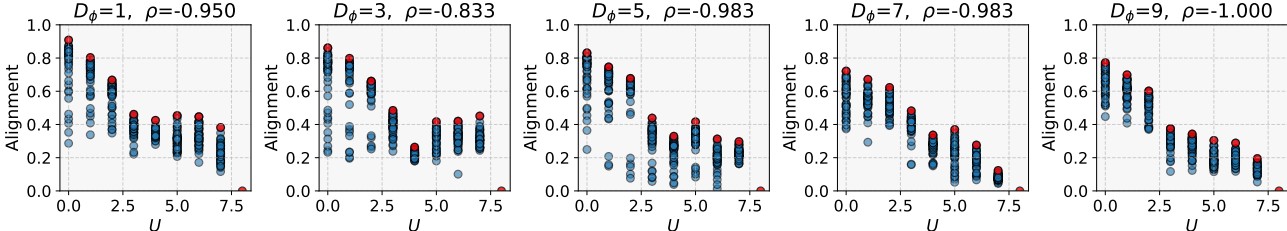

Figure 4: **Alignment vs uniqueness on synthetic datasets.** Alignment is computed between unimodal encoders trained on datasets with different levels of informational uniqueness $U$. Each dot is an independent run on a different model size on a given dataset. We see that the maximum level of achievable level of alignment, shown by the red dots, decreases as the level of uniqueness increases. Five different figures shows the different levels of nonlinear transformation we apply to the original data. We report the Spearman correlation $\rho$ between the maximum alignment values and $U$.

the Wikipedia caption dataset. Using a pretrained sentence BERT model to quantify semantic similarity between the original caption and the GPT-4 captions, we verify that the average semantic similarity monotonically decreases as the level of uniqueness increases. See Appendix C.2 for more details.

We experiment with MultiBench (Liang et al., 2021) which collects a diverse range of real-world multimodal datasets: MOSEI (Bagher Zadeh et al., 2018), a dataset for predicting emotions from videos (vision, audio, language); MOSI (Zadeh et al., 2016), a dataset for predicting sentiment from videos (vision, audio, language), URFUNNY (Hasan et al., 2019), a humor detection dataset from videos (vision, audio, language); MUStARD (Castro et al., 2019), a sarcasm detection dataset from TV shows (vision, audio, language); and AVMNIST (Pérez-Rúa et al., 2019), a dataset for digit classification from paired images and spoken digits. Additionally, we experiment with MM-IMDb (Arevalo et al., 2017), a dataset for classifying movie genres from paired images and text. While we cannot explicitly vary the information content, past work has collected human annotations of the levels of redundancy and uniqueness in these datasets, showing that most multimodal datasets have a significant amount of uniqueness (Liang et al., 2023a).

**Computing Alignment.** For models trained on synthetic data, we evaluate alignment using unbiased Centered Kernel Alignment (CKA) (Kornblith et al., 2019). See Appendix D.1 for results with additional metrics. Following the methodology outlined in Huh et al. (2024), for large pretrained vision and language models, we evaluate alignment using mutual KNN, a variant of CKA. See Appendix B for more details.

## 4. RQ1: When does Alignment Emerge?

We empirically evaluate whether alignment emerges naturally by systematically varying redundancy, uniqueness, and

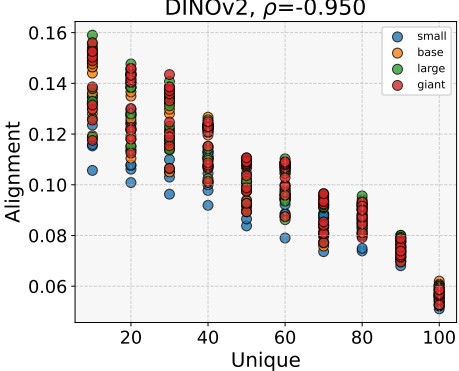

Figure 5: **Alignment vs uniqueness on real large-scale vision-language datasets**. Alignment is computed between DINOv2 vision models and large language models. Each dot is an independent run on a different model size on a dataset with a given level of uniqueness. The maximum achievable alignment decreases as uniqueness increases.

heterogeneity. In the synthetic setting, the level of uniqueness $U$ denotes the number of unique features used in computing the label. In Figure 4, we observe that as $U$ increases, the maximum alignment decreases across different model depths and transformation depths. A similar trend is evident in Figure 5, which examines the alignment between large-scale language models and DINOv2 (Oquab et al., 2023) vision models over different levels of $U$ is the percentage of perturbation. See Appendix D.5 for experiments with more vision models. An additional experiment, detailed in Appendix D.1, demonstrates that data heterogeneity is negatively correlated with the level of achievable alignment. Collectively, these experiments provide strong empirical evidence supporting the hypothesis that the level of alignment is indeed constrained by the degrees of heterogeneity and interactions between the modalities.

We now investigate whether increasing model capacity can improve alignment between representations of increasingly

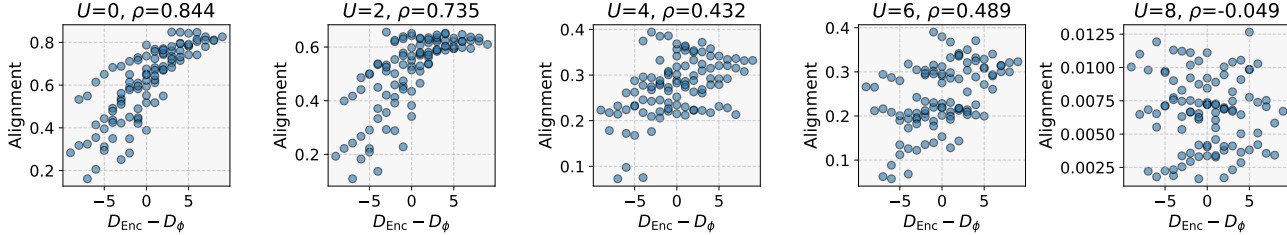

Figure 6: **Emergence of alignment across heterogeneity and uniqueness.** We plot alignment with respect to $(D_{Enc} - D_\phi)$ for different levels of uniqueness and report the Spearman correlation $\rho$. When redundancy is high, we see that alignment emerges when $(D_{Enc} - D_\phi)$ is high. However, as uniqueness increases, the correlation between $(D_{Enc} - D_\phi)$ and alignment vanishes.

heterogeneous and unique modalities. In Figure 6, we plot alignment scores as a function of $(D_{Enc}, D_\phi)$, where $D_{Enc}$ represents the encoder depth and $D_\phi$ represents the transformation depth of the second modality. When uniqueness is low, we observe that alignment improves significantly when the model capacity (relative to the transformation depth) is greater. This suggests that increased model capacity is effective in handling heterogeneity between modalities. Concretely, in these scenarios, alignment appears to follow the trend $(D_{Enc} - D_\phi) \propto$ Alignment, meaning that the relative capacity of the encoder compensates for the complexity introduced by the transformation depth. However, as uniqueness increases, the relationship between alignment and relative model capacity becomes much weaker. In these cases, $(D_{Enc} - D_\phi)$ no longer predicts higher alignment scores. This indicates that when modalities have a high level of unique information, simply increasing model capacity is insufficient to achieve higher alignment. Instead, other factors—such as the degree of shared information—may become the limiting factor in determining alignment.

In summary, while model size and capacity are correlated with alignment, there exists an upper limit to the level of achievable alignment, which is fundamentally determined by the intrinsic properties of the data. This finding implies that perfect alignment cannot be simultaneously achieved with optimal performance when the data modalities inherently differ in their information content. Moreover, increasing model depth only effectively aligns heterogeneous modalities when they contain highly redundant information and can fail for high uniqueness.

## 5. RQ2: Is Alignment Correlated with Performance?

In this section, we systematically investigate the relationship between alignment and performance, identifying scenarios where alignment enhances performance and others where it may introduce unintended trade-offs.

### 5.1. Alignment-performance vs interactions/uniqueness

For each synthetic dataset, we analyze the relationship between alignment, performance, and model capacity. We include model capacity in our analysis as increased capacity generally leads to better performance and is assumed to correlate with better alignment. Our findings are summarized in Figure 7, where we plot the correlations between alignment and performance across different dataset dimensions, where $U$ is defined in Section 4. In highly redundant settings, the correlation between alignment and performance is strong, with relatively little variation across different levels of heterogeneity and random seeds. However, as uniqueness increases, the median correlation decreases toward zero, and the range of correlations expands significantly. Notably, for $U > 3$, the correlation even becomes negative in some cases, suggesting that higher uniqueness can disrupt the relationship between alignment and performance. A similar trend is observed when examining the correlation between alignment and model depth in Figure 7 (right). As uniqueness increases, both the median correlation decreases and the variance in correlation increases substantially, with instances of anticorrelated alignment-depth relationships. While deeper models do not necessarily lead to better alignment when uniqueness is high, we see in Figure 7 (center) that performance and depth remain positively correlated across different levels of uniqueness, with much lower variance in correlation at higher uniqueness levels. This suggests that while alignment may not always be a reliable predictor of performance, increasing model capacity can still improve task performance.

We next verify whether these findings extend to large vision-language models. In Figure 8, we compute linear fits to alignment to DINOv2 and language model performance. As uniqueness increases, the slope of the linear fit decreases, showing that the relation between cross-modal alignment and performance weakens. Nevertheless, we see that the better performing language models are those with greater capacity, showing that increasing capacity can lead to better performance even when alignment does not emerge. We include more analysis in Appendix D.6 involving different

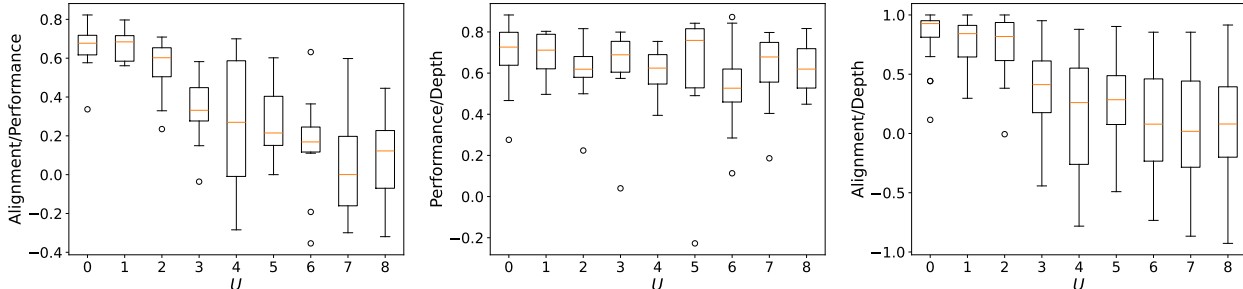

Figure 7: **Alignment, performance, and depth correlation plots across different synthetic depths and experiment seeds.** In each plot, we show the spread of Spearman correlation coefficients $\rho$ for each level of uniqueness, where the orange lines are the median correlations and the dots are outliers. **Left**: When the two modalities are fully redundant, the alignment is strongly correlated with performance. When the two modalities have high uniqueness, alignment has a vanishing correlation with performance. In fact, for a significant proportion of tasks, the correlation is negative. **Mid**: In contrast, model size measured by depth always has a strong positive correlation with performance and does not seem to change across datasets. This means that representation alignment may not be a universal phenomenon, and is introduced by some special properties of data. In contrast, the influence of model size on performance seems universal and is consistent with the well-observed scaling laws. **Right**: For each level of uniqueness, we show the variance in alignment/depth correlation. As uniqueness increases, the median alignment/depth decreases to 0, and the range of correlation values increases significantly.

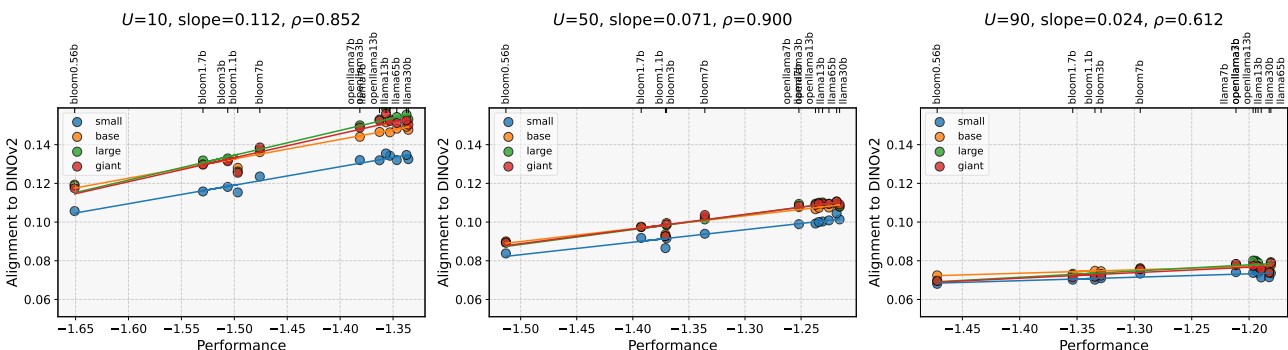

Figure 8: **Alignment vs performance across different uniqueness.** We plot the vision-language alignment using DINOv2 vision models with respect to language model performance, measured using negative `bits-per-byte-loss`. We show individual best fit lines for each size of vision model and the average Spearman correlation coefficient $\rho$. As $U$ increases, the slope of the linear fit decreases, showing that better performance can be achieved without increased alignment.

vision model training schemes. Overall, our experiments on both synthetic data and large vision-language models indicate that as uniqueness increases, higher-performing models with greater capacity do not necessarily exhibit stronger alignment. This reinforces the conclusion that alignment does not always predict model effectiveness, particularly when the modalities contain significant amounts of unique information.

## 5.2. Alignment-performance vs heterogeneity

Additionally, we analyze whether alignment correlates with performance across varying levels of heterogeneity in Figure 9. Intuitively, we expect higher levels of heterogeneity to result in lower performance, but it is unclear whether this trend is reflected in alignment scores. Our findings

show that while alignment and performance exhibit a strong linear relationship at low levels of uniqueness, this relationship weakens as uniqueness increases. Specifically, with higher uniqueness, models trained on similar modalities do not consistently achieve better alignment than those trained on heterogeneous modalities. This suggests that alignment does not uniformly degrade with increasing heterogeneity and that the interaction between uniqueness, heterogeneity, and alignment is more complex than a simple linear relationship. These results further reinforce the idea that alignment alone is not a sufficient predictor of model performance, especially in multimodal settings where modalities contain varying levels of interactions and heterogeneity.

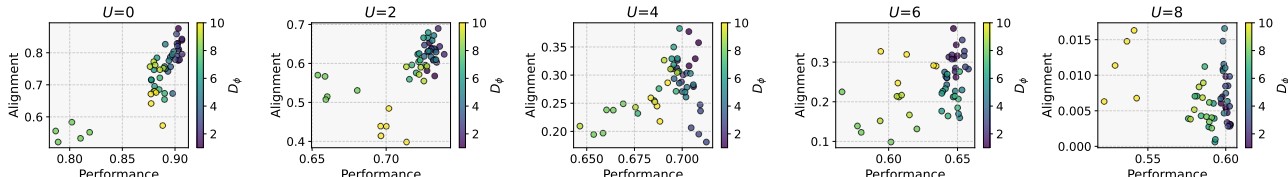

Figure 9: **Alignment vs performance across levels of heterogeneity.** We plot the alignment and performance scores at different levels of heterogeneity, with the transformed modality's encoder fixed to the maximum transformation depth. At high levels of uniqueness, we see that high performance correlates with high alignment, with both being greater at lower synthetic depths. Past $U = 4$, we see that while performance is higher at lower synthetic depths, the alignment scores on these datasets are not necessarily higher.

| **Dataset** | **Vision - Audio** | | **Vision - Text** | | **Audio - Text** | |
|---|---|---|---|---|---|---|
| | Vision | Audio | Vision | Text | Audio | Text |
| MOSEI (Bagher Zadeh et al., 2018) | -0.223 | -0.172 | -0.093 | -0.482 | -0.145 | -0.641 |
| MOSI (Zadeh et al., 2016) | -0.014 | 0.313 | 0.116 | -0.413 | 0.403 | -0.386 |
| URFUNNY (Hasan et al., 2019) | -0.332 | -0.336 | -0.232 | 0.284 | -0.373 | 0.083 |
| MUStARD (Castro et al., 2019) | 0.381 | 0.208 | 0.436 | 0.019 | 0.216 | 0.438 |
| AVMNIST (Pérez-Rúa et al., 2019) | 0.887 | 0.723 | - | - | - | - |

Table 1: **Alignment-performance correlations on MultiBench.** We compute the correlation between model performance and alignment across 4 affective computing datasets with tasks that require unique information in vision, audio, and language modalities. We additionally benchmark on AVMNIST, a dataset with high redundancy as the modalities are images of digits and spoken digits for digit classification. On the affective computing datasets, the correlation is weak and often negative, suggesting that enforcing alignment between modalities may not be desirable. In contrast, the alignment of vision and audio modalities in AVMNIST is highly correlated with performance.

## 6. RQ3: Alignment-Performance Correlation is an Inherent Property of Datasets

Finally, we investigate how the alignment-performance correlation varies across real-world multimodal datasets. Quantifying this relation is important to practitioners, as a positive alignment-performance correlation suggests that a practitioner can improve performance by explicitly aligning modalities. As shown in our experiments in Section 5, we expect that on tasks involving redundant information, alignment positively correlates with performance whereas for tasks that require unique information in modalities, the correlation may be weaker and not necessarily positive.

### 6.1. MultiBench Datasets

We evaluate these hypotheses on a subset of datasets from MultiBench (Liang et al., 2021) with varying degrees of task-relevant redundant and unique information content, including MOSEI (Bagher Zadeh et al., 2018), a dataset for predicting emotions from videos (vision, audio, text); MOSI (Zadeh et al., 2016), a dataset for predicting sentiment from videos (vision, audio, text), URFUNNY (Hasan et al., 2019), a humor detection dataset from videos (vision, audio, text); MUSTARD (Castro et al., 2019), a sarcasm detection dataset from TV shows (vision, audio, text); and AVMNIST (Pérez-Rúa et al., 2019), a dataset for digit classification from paired images and spoken digits (vision,

audio). See Appendix C.3 for details about the datasets. We train transformers with varying depths for each modality and compute the cross-modal alignment. See Appendix A for details on our experiment setup.

We show these results in Table 1. On sentiment analysis tasks that typically require unique information from language, alignment, and performance are weakly correlated or even negatively correlated. For a given dataset, the alignment-performance relationship can even vary between different modalities. For example, on MUStARD, alignment is more highly correlated with vision performance, whereas audio and text performance do not seem as correlated. On AVMNIST, alignment strongly correlates with performance for both modalities, as the information content is largely redundant information about the digit identity. These results corroborate our findings that the alignment-performance relationship heavily depends on dataset characteristics.

### 6.2. Algorithmic use case for quantifying alignment-performance relation

To show how quantifying the alignment-performance relation can impact algorithm design, we consider a practical setting where there is a large dataset of paired input data, but only a small subset of the dataset has labels due to the cost of annotation. One approach to leverage the unlabeled paired data is to finetune a pretrained model using both a su-

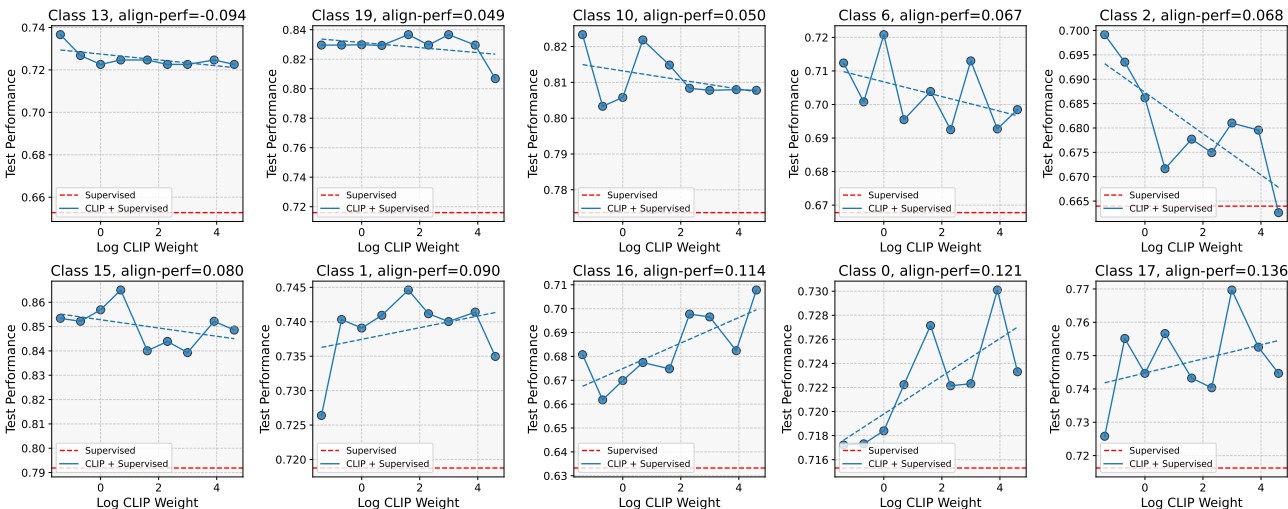

Figure 10: **Alignment-performance relation predicts impact of explicit alignment on downstream performance on MM-IMDb.** We finetune CLIP vision and language encoders on MM-IMDb using $\mathcal{L} = \mathcal{L}_{sup} + w * \mathcal{L}_{\text{CLIP}}$, where $\mathcal{L}_{sup}$ is the supervised loss computed on top of the language embeddings and the contribution of $\mathcal{L}_{\text{CLIP}}$ is modulated by a weight $w$. **Left-to-right, top-to-bottom:** The classification tasks are ordered by increasing alignment-performance linear fit slopes, which are computed using unimodal language and vision models. The dotted blue line shows the linear fit to performance across varying degrees of explicit alignment, and the dotted red line shows that finetuning with only $\mathcal{L}_{sup}$ results in overfitting. Results demonstrate that alignment-performance relations predict how the amount of explicit alignment (controlled by $w$) impacts performance. Specifically, classes 13, 19, 10, 6, 2, 15 have smaller alignment-performance linear fit slopes, which correspond to a weak or negative relation between $w$ and performance. In contrast, classes 1, 16, 0, and 17 have higher alignment-performance linear fit slopes, in which increasing $w$ generally improves performance.

pervised loss and an explicit alignment objective, such as the CLIP loss: $\mathcal{L} = \mathcal{L}_{sup} + w * \mathcal{L}_{\text{CLIP}}$. However, this raises the question of how the contribution of these two losses should be balanced to maximize performance. From our analysis, we posit that the "ideal" amount of alignment is dataset and task-specific. If the alignment-performance relation is weak, then performance degrades or does not change when increasing the weight on the explicit alignment objective. Conversely, when the alignment-performance relation is stronger, performance should increase with a larger weight on the alignment objective.

To test this hypothesis, we run experiments on MM-IMDb (Arevalo et al., 2017), a realistic dataset for classifying movie genres from movie posters and text descriptions of the movie plot, where there are 23 classes. As such, the multilabel classification task can be broken down into 23 binary classification tasks. See Appendix D.7 for the distribution of alignment-performance linear fit slopes. Compared to the original dataset size of 25k examples, we use a subset of 1024 labeled examples for each of the train, validation, and test sets to simulate the scarce data scenario. For 10 different classes, we compute cross-modal alignment between the same vision models and language models as Huh et al. (2024), and downstream performance is measured by training a linear classifier on the language embeddings. We finetune CLIP vision and language models with $\mathcal{L}$, where

$w \in \{0, 0.1, 0.25, 0.5, 1.0, 2.0, 5.0, 10.0, 50.0, 100.0\}$. In agreement with our analysis, we demonstrate in Figure 10 that on the categories with lower alignment-performance slopes, increasing $w$ leads to worse performance, whereas for classes with higher alignment-performance slopes, high values of $w$ improve performance. These results demonstrate the use of quantifying the relation between alignment and performance, even with unimodal models that are not explicitly aligned, for deciding how much to explicitly align the modalities.

## 7. Conclusion

This paper provides a comprehensive analysis of the relationship between multimodal alignment, performance, and multimodal data characteristics. We offer a nuanced perspective on how alignment emerges across different modalities and how its effectiveness is influenced by the interactions and heterogeneity within the data. Specifically, our findings show that as uniqueness and heterogeneity increase, the emergence of alignment weakens, and that alignment often fails to track performance in datasets with higher uniqueness. In the case of perfect redundancy, our result supports the Platonic Representation Hypothesis, but as the amount of unique information and data heterogeneity increases, our results provide a generalization of this phenomenon.

# Acknowledgements

MT is supported by the National Science Foundation (NSF) under Grant No. 2141064. We acknowledge NVIDIA's GPU support. We thank Hengzhi Li and Minseok Jung for feedback and discussions.

# Impact Statement

This paper presents an empirical analysis whose goal is to advance the field of multimodal representation learning. Specifically, our work opens up the possibility of characterizing and quantifying multimodal datasets via alignment-performance relationships. This can help advance our understanding of multimodal data and inspire the design of better methods that appropriately align (or perhaps even unalign) modality representations when necessary. Our work also inspires new theoretical questions regarding why different models sometimes converge to similar representations, even though they are often overparametrized and theoretically capable of learning arbitrary representations. Answering these questions can advance our understanding of today's large-scale multimodal AI systems. Multimodal models are broadly impactful for many real-world applications including understanding human verbal and nonverbal communication, fusing multiple physical sensors, and analyzing multiple sources of medical data.

There are many potential long-term societal consequences of our work, but since our work is primarily an empirical analysis, we feel that none of these impacts must be specifically highlighted here.

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

# A. Experimental Details

## A.1. Synthetic Data Experiments

On the synthetic dataset, we train MLPs with the AdamW optimizer with the number of hidden dimensions kept the same as the number of input features, 12. For a given level of uniqueuess, we choose suitable hyperparameters across different model depths and transformation depths. Specifically, we tune the learning rate in the range $\{1e-1, 1e-2, 1e-3, 1e-4\}$ and weight decay in the range $\{0, 1e-1, 1e-2, 1e-3, 1e-4\}$ for each modality. The depth 1 MLP for the untransformed modality were trained for 50 epochs and the models for the transformed modality were trained for 300 epochs. We use a batch size of 512 for computing alignment. To ensure robustness, we report results with five different random seeds for each dataset.

## A.2. Vision-Language Alignment

We evaluate alignment using the same set of language and vision models as Huh et al. (2024). The language model families considered are BLOOM (Workshop et al., 2023), OpenLLaMA (Geng & Liu, 2023), and LLaMA (Touvron et al., 2023) downloaded from HuggingFace (Wolf et al., 2020). The vision models are vision transformer models of various sizes trained on various data and objectives. These include classification on ImageNet-21K (Russakovsky et al., 2015), MAE (He et al., 2022), DINOv2 (Oquab et al., 2023), CLIP (Radford et al., 2021b), and CLIP finetuned on ImageNet-12K. These models were downloaded from PyTorch Image Models (Wightman, 2019).

## A.3. MultiBench Experiments

We train transformers on the pre-extracted video, audio, and text features for the affective computing datasets and the audio modality of AVMNIST, and vision transformers on the AVMNIST digit images. For each modality, we vary the depth of the transformers in the range $\{1, \ldots, 10\}$. We use a single head for self-attention and set the embedding size to the input dimension. For classification tasks, we append a `[cls]` token to the sequence with a learnable embedding. The embedding of this token is used to compute alignment between layers; otherwise, we do average pooling over the input sequence. We use the AdamW optimizer. For each dataset, we choose suitable hyperparameters across different model depths and tune the learning rate in the range $\{1e-3, 5e-4, 1e-4, 5e-5, 1e-5\}$ and and weight decay in the range $\{0, 1e-1, 1e-2, 1e-3, 1e-4\}$.

To ensure robustness, we train each architecture across 3 different seeds, providing 30 alignment-performance data points. To get the alignment-performance correlation given modalities 1 and 2, the alignment of every modality 2 model is computed with respect to the modality 1 model with the highest validation score across the different seeds, and we report the correlation of these alignment scores with the performance of the modality 2 models.

## A.4. MM-IMDb Experiments

To compute downstream performance, we train linear classifiers on top of the final hidden layer embeddings of the language model described in Appendix A.2 for 100 epochs. We tune the learning rate in the range $\{5e-3, 1e-3, 5e-4, 1e-4\}$ and weight decay in the range $\{0, 1e-1, 1e-2, 1e-3, 1e-4\}$. For finetuning models trained with CLIP, we use a learning rate of $10^{-4}$ and a cosine scheduler with final value of $10^{-6}$ and a warmup over 10 epochs. Models were optimized for 30 epochs.

# B. Alignment Computation

Given mean-centered feature sets of $n$ samples, $Z_1, Z_2 \in \mathbb{R}^{n \times d}$, from two modalities $X_1$ and $X_2$, we first compute the covariances of these two different feature sets, and then compute the empirical estimator of the Hilbert-Schmidt Independence Criterion (Gretton et al., 2005) using a linear kernel. Hence,

$$\mathsf{HSIC}(Z_1 Z_1^T, Z_2 Z_2^T) = \frac{1}{(n-1)^2} \mathrm{Tr}(Z_1 Z_1^T Z_2 Z_2^T) = \frac{1}{(n-1)^2} \|Z_1^T Z_2\|_F^2 \tag{1}$$

The Centered Kernel Alignment (CKA) (Kornblith et al., 2019) is then obtained by normalizing HSIC to ensure scale invariance and comparability across different feature sets:

$$\mathrm{CKA}(Z_1, Z_2) = \frac{\mathrm{HSIC}(Z_1 Z_1^T, Z_2 Z_2^T)}{\sqrt{\mathrm{HSIC}(Z_1 Z_1^T, Z_1 Z_1^T)\mathrm{HSIC}(Z_2 Z_2^T, Z_2 Z_2^T)}} \tag{2}$$

As demonstrated in (Huh et al., 2024), the definition of alignment can be adjusted to limit the cross-covariance measurement to only those samples identified as nearest neighbors of the current sample $i$. This modification prioritizes similarity over dissimilarity, thereby emphasizing local alignment:

$$\mathrm{ALIGN_{MKNN}}(Z_1, Z_2) = \sum_i \sum_j \alpha(i, j)) \tag{3}$$

$$\text{where, } \alpha(i, j) = \mathbf{1}[Z_{1,j} \in knn(Z_{1,i}) \wedge Z_{2,j} \in knn(Z_{2,i}) \wedge i \neq j] \tag{4}$$

Here, $Z_{1,k}$ and $Z_{2,k}$ refer to the $k^{\text{th}}$ row of $Z_1$ and $Z_2$, respectively, while MKNN denotes Mutual KNN.

Thus, Mutual-KNN MKNN is defined as:

$$\mathrm{MKNN}(Z_1, Z_2) = \frac{\mathrm{ALIGN_{MKNN}}(Z_1, Z_2)}{\sqrt{\mathrm{ALIGN_{MKNN}}(Z_1, Z_1) \cdot \mathrm{ALIGN_{MKNN}}(Z_2, Z_2)}} \tag{5}$$

Following Huh et al. (2024), we use $k = 10$ nearest neighbors over 1024 samples from the Wikipedia caption dataset. For the vision model, the class token of each layer is used, and for the language model, the embeddings of a given layer are average pooled to a single token. $l_2$ normalization is applied to the features and elements in the features that are above the 95-th percentile are truncated.

After computing the alignment between all pairs of layers between $E_1$ and $E_{2,d}$ using CKA or mutual KNN, we report the best alignment score across all layer pairs (Schrimpf et al., 2018).

## C. Dataset Details

### C.1. Synthetic Data

We discuss in detail how we construct a synthetic dataset with two modalities to analyze how uniqueness, redundancy, and heterogeneity influence the emergence of alignment. Let $x_1 = [x_r, x_{u_1}]$ and $x_2 = [x_r, x_{u_2}]$. Here, $x_r \in \mathbb{R}^{n_R}$ represents the redundant information shared between the two modalities, while $x_{u_1}, x_{u_2} \in \mathbb{R}^{n_U}$ denote the unique information for each modality. Both $x_1$ and $x_2$ represent arbitrary data samples.

For each data sample, we generate $x_r$, $x_{u_1}$, and $x_{u_2}$ by sampling binary vectors from a uniform distribution. Specifically, $x_r \sim \mathrm{Uniform}(\{0,1\}^{n_R}), x_{u_1} \sim \mathrm{Uniform}(\{0,1\}^{n_U})$, and $x_{u_2} \sim \mathrm{Uniform}(\{0,1\}^{n_U})$.

To define the labels for this dataset, we introduce task masks $M_R \in \mathbb{R}^{n_R}$ and $M_{U_1}, M_{U_2} \in \mathbb{R}^{n_U}$, which determine the features used in computing the output labels. These masks indicate whether a particular feature contributes to the label-generation process. Specifically, the task masks are defined as follows, where the subscript $i$ refers to the $i^{\text{th}}$ entry of the respective mask vector:

$$M_{R_i} = \begin{cases} 1 & \text{if } 0 \leqslant i < n_R, \\ 0 & \text{otherwise.} \end{cases} \tag{6}$$

$$M_{U_{1_i}} = \begin{cases} 1 & \text{if } 0 \leqslant i < \lceil \frac{n_U}{2} \rceil, \\ 0 & \text{otherwise.} \end{cases} \tag{7}$$

$$M_{U_{2_i}} = \begin{cases} 1 & \text{if } 0 \leqslant i < \lfloor \frac{n_U}{2} \rfloor, \\ 0 & \text{otherwise.} \end{cases} \tag{8}$$

We define the task label $y$ as a function $\psi_Y(\cdot)$ of the masked components $x_r \odot M_R$, $x_{u_1} \odot M_{U_1}$, and $x_{u_2} \odot M_{U_2}$, such that:

$$y = \psi_Y(x_r \odot M_R, x_{u_1} \odot M_{U_1}, x_{u_2} \odot M_{U_2}), \tag{9}$$

where $x_r \odot M_R$ captures the task-relevant redundant information, $x_{u_1} \odot M_{U_1}$ captures the task-relevant unique information from modality 1 and $x_{u_2} \odot M_{U_2}$ captures the task-relevant unique information from modality 2.

Here, $\odot$ denotes the element-wise (Hadamard) product. Intuitively, the task masks $M_R$, $M_{U_1}$, and $M_{U_2}$ are essential for controlling the relative contributions of the redundant ($x_r$) and unique ($x_{u_1}$, $x_{u_2}$) components to the label generation process.

In our synthetic experiments, we assume the joint distribution of the components as follows:

$$\mathbb{P}(X_C, X_{U_1}, X_{U_2}) = \mathbb{P}(X_C)\mathbb{P}(X_{U_1})\mathbb{P}(X_{U_2}) \tag{10}$$
$$\text{Where, } \mathbb{P}(X_C) = \text{Uniform}(\{0,1\}^{n_R}) \tag{11}$$
$$\mathbb{P}(X_{U_1}) = \text{Uniform}(\{0,1\}^{n_U}) \tag{12}$$
$$\mathbb{P}(X_{U_2}) = \text{Uniform}(\{0,1\}^{n_U}) \tag{13}$$

This formulation assumes that the redundant information ($x_r$) and the unique components ($x_{u_1}$, $x_{u_2}$) are all independently distributed.

The task masks play a critical role in modulating which features are used to compute the labels, thereby allowing precise control over the relative importance of shared and unique information in the synthetic dataset. This design facilitates the study of how these components influence alignment and downstream task performance.

In our experiments, we fix $n_Y$ (the number of features relevant for the task) while varying $n_R$, $n_U$ to explore different dataset configurations. Concretely, $n_Y = n_R + n_U$.

When $n_y = n_R$, both modalities have equal amounts of task-relevant information, allowing them to perform equally well on the downstream classification task. However, when $n_R < n_Y$, the shared information $x_c$ becomes insufficient to fully capture the task-relevant features. By adjusting the proportion of $\frac{n_R}{n_Y}$, we heuristically vary the amount of task-specific shared information. This allows us to explore how the balance of redundant and unique information impacts alignment and downstream performance.

An example of this is when $n_Y = 2$, $x_r \sim \text{Uniform}(\{0,1\})$, $x_{u_1} \sim \text{Uniform}(\{0,1\})$, $x_{u_2} \sim \text{Uniform}(\{0,1\})$. Given that the label function is an OR function of $[x_r, x_{u_1}]$, where $n_U = 1$, we can see how the label predictions would differ based on $x_1 = [x_r, x_{u_1}]$ and $x_2 = [x_r, x_{u_2}]$ in Table 2.

| $x_1$ | $x_2$ | $\hat{y}_1$ | $\hat{y}_2$ | y |
|-------|-------|-------------|-------------|---|
| 00 | 00 | 0 | 0 | 0 |
| 01 | 00 | 1 | 0 | 1 |
| 00 | 01 | 0 | 0 | 0 |
| 01 | 01 | 1 | 0 | 1 |
| 10 | 10 | 1 | 1 | 1 |
| 11 | 10 | 1 | 1 | 1 |
| 10 | 11 | 1 | 1 | 1 |
| 11 | 11 | 1 | 1 | 1 |

Table 2: $n_U = 1$ **Predictions.** $\hat{y}_2$ is incorrect for 2 examples due to lacking the unique information.

In contrast, when the labels are an OR of $[x_{u_1}, x_{u_2}]$, where $n_U = 2$, we can see how the label predictions would differ based on $x_1 = [x_r, x_{u_1}]$ and $x_2 = [x_r, x_{u_2}]$ in Table 3.

To incorporate heterogeneity into the setup, we transform the second modality ($x_2$) using a nonlinear function $\phi(\cdot)$. Specifically, $\phi(\cdot)$ is modeled as a multilayer perceptron (MLP), where the number of layers ($D_\phi$), also referred to as transformation depth, serves as a heuristic measure of heterogeneity. A higher $D_\phi$ implies a more complex transformation, thereby increasing the heterogeneity between the two modalities. Hence, concretely, $x_{2,\phi} = \phi([x_c, x_{u_2}])$.

In all our experiments, we fix $n_Y = 8$, which represents the total number of task-relevant features. We vary $n_R \in \{0, \ldots, 8\}$,

| $x_1$ | $x_2$ | $\hat{y}_1$ | $\hat{y}_2$ | y |
|-------|-------|-------------|-------------|---|
| 00 | 00 | 0 | 0 | 0 |
| 01 | 00 | 1 | 0 | 1 |
| 00 | 01 | 0 | 1 | 1 |
| 01 | 01 | 1 | 1 | 1 |
| 10 | 10 | 0 | 0 | 0 |
| 11 | 10 | 1 | 0 | 1 |
| 10 | 11 | 0 | 1 | 1 |
| 11 | 11 | 1 | 1 | 1 |

Table 3: $n_U = 2$ **Predictions.** Both $\hat{y}_1$ and $\hat{y}_2$ are incorrect for 2 examples due to lacking unique information in the other modality.

thereby controlling the amount of redundant (shared) task-specific information. Consequently, the amount of unique task-specific information is determined as $n_U = n_Y - n_R$. We refer to the level of unique information as $U = n_U$.

### C.2. Wikipedia-Image Text Dataset with Uniqueness

Below is the prompt to GPT-4o to create captions with unique information.

---

**Annotation Instructions**

Imagine you have been assigned the task of progressively enhancing the following caption by systematically introducing unique and differentiating details:
**Original Caption:** <caption>
### **Task Overview:** You will generate **10 increasingly different variations** of this caption, ensuring that each version changes the semantic meaning of the **original caption**. If an image is provided, ensure that the changes to the caption are semanticaly different **distinct from the visual elements in the image**.
### **Definition of Changing Semantic Meaning** Changing semantic meaning means that the modified caption should **alter the image if used in a generation model**.
This can be achieved by changing visual cues of the original caption including but not limited to:
– Identity of objects or people
– Textures of objects or landscape elements
– Location, time of day, weather, or environment specifics
### **Task Breakdown & Structure:**
1. **Incremental Enhancement:**
    – Generate **10 versions** of the caption.
    – Each version should introduce an increasing amount of semantic differences by increments of **[10, 20, ..., 100] (in percentage)**.
2. **Gradual Transformation:**
    – Ensure that each step logically builds upon the previous one.
    – The final version should have a completely different semantic meaning from the original caption.
3. **Handling Image Input (if provided):**
    – If an image is provided, ensure that **the semantics of the changed captions are different from the visual elements in the image**.
4. **Output Formatting:**
    – Each caption should be **separated by a consistent delimiter** to ensure clarity.
    – Use the following format for **each generated caption:**
    – Caption N - Uniqueness Percentage%: Generated Caption
    – Ensure that each step **logically evolves** from the previous version, creating a seamless and natural transformation.
### **Expected Output Format Example:**
*Input Caption*: Golden hues gently stretch across the horizon, deepening as the sun slowly dips, casting soft amber

---

reflections on the tranquil sea.
Caption 1 - 10%: Crimson and violet hues gently stretch across the horizon, deepening as the sun slowly dips, casting reflections on the tranquil sea.
Caption 2 - 20%: Crimson and violet hues gently stretch across the horizon, deepening as the sun slowly dips, casting reflections on the waves.
Caption 3 - 30%: Crimson and violet hues gently stretch across the horizon, deepening as the sun rises, casting reflections on the waves.
<Captions 4-10 omitted for brevity>
### **Goal:** By the end, the series of 10 captions should **illustrate a clear evolution** in semantic meaning both in terms of text and any provided image.
—
**Input Parameters:**
– **Caption:** "caption"
– **(Optional) Image:** A visual reference that must also be considered when introducing unique details.
Your task is to ensure that each new version would generate an image that is **perceptibly different** from both the original caption and any provided visual input.

## C.3. MultiBench Dataset

Below, we discuss the MultiBench datasets in more detail.

- **MUStARD** (Castro et al., 2019) is a dataset for automated sarcasm discovery, compiled from popular TV shows, including Friends, The Golden Girls, The Big Bang Theory, and Sarcasmaholics Anonymous. There are 414, 138, and 138 video segments in the training, validation, and testing data, which gives a total of 690 data points.

- **MOSI** (Zadeh et al., 2016) is a dataset for sentiment analysis consisting of 2,199 opinion video clips. Each video is further split into short segments (roughly 10-20 seconds) that are annotated, resulting in 1284, 229, 686 segments in the train, validation, and testing sets. As the annotations are sentiment intensity, which ranges from [-3, 3], we train our models on the continuous labels with L1 loss and evaluate positive-negative classification accuracy.

- **UR-FUNNY** (Hasan et al., 2019) is a large-scale dataset for humor detection in human speech, consisting of more than 16000 video samples ( from TED talks collected from 1866 videos. There are a total of 10,598, 2,626, and 3,290 segments in the train, validation, and testing sets. Humor is annotated as either positive or negative, with a homogeneous 50% split in the dataset.

- **MOSEI** (Bagher Zadeh et al., 2018) is a large-scale dataset for sentence-level sentiment analysis and emotion recognition from real-world online videos, containing more than 65 hours of annotated video from more than 1,000 speakers and 250 topics. There are a total of 16,265, 1,869, and 4,643 segments in the train, validation, and testing sets, resulting in 22,777 data points. As in MOSI, we train our models on continuous sentiment intensity labels with L1 loss and evaluate positive-negative classification.

- **AVMNIST** (Pérez-Rúa et al., 2019) is a dataset created by pairing audio of human reading digits from the FSDD dataset (Jackson, 2025) with written digits in the MNIST dataset (Lecun et al., 1998) with a task to predict the digit into one of 10 classes (0-9). While common practice (Pérez-Rúa et al., 2019) is to increase the difficulty by removing 75% of energy in the visual modality via PCA and adding noise from ESC-50 to the audio modality, we use the unnoised image and audio modalities in order to preserve the redundant information between modalities. An audio sample from FSDD with matching digit identity is paired with each image in MNIST, resulting in 55000, 5000, and 10000 examples in the train, validation, and test sets respectively. We train vision transformers on MNIST images that are converted to 4x4 patches with a sequence length of 49. We preprocess the raw FSDD audio into 36 MFCC coefficients with a maximum sequence length of 20 using librosa (McFee et al., 2015).

## C.4. MM-IMDb Dataset

Multimodal IMDb (Arevalo et al., 2017) is a large-scale real world dataset. It is curated by filtering out movies from the Movielens 20M dataset that lack a poster. Each data point in MM-IMDb consists of the movie poster and plot summary, as

well as additional metadata such as year, language, director, etc. In our experiments, we use the raw data instead of the preprocessed features from Multibench. In addition, we consider classifying different movie genre as separate downstream tasks and compute alignment-performance linear fits for each genre.

# D. Additional Figures

## D.1. Synthetic Data Results with Different Alignment Metrics

In Figures 11, 12, and 13, we plot the alignment between unimodal encoders with respect to uniqueness using different alignment metrics, including unbiased CKA (Kornblith et al., 2019) with linear and RBF kernels, SVCCA (Raghu et al., 2017), mutual $k$-NN (Huh et al., 2024)), as well as with different batch sizes.

In Figures 14, 15, and 16, we plot the alignment between unimodal encoders with respect to heterogeneity. In Figures 17, 18, and 19, we plot the alignment, performance, and depth correlations using different alignment metrics and batch sizes. Overall, our results are consistent across various alignment metrics and batch sizes.

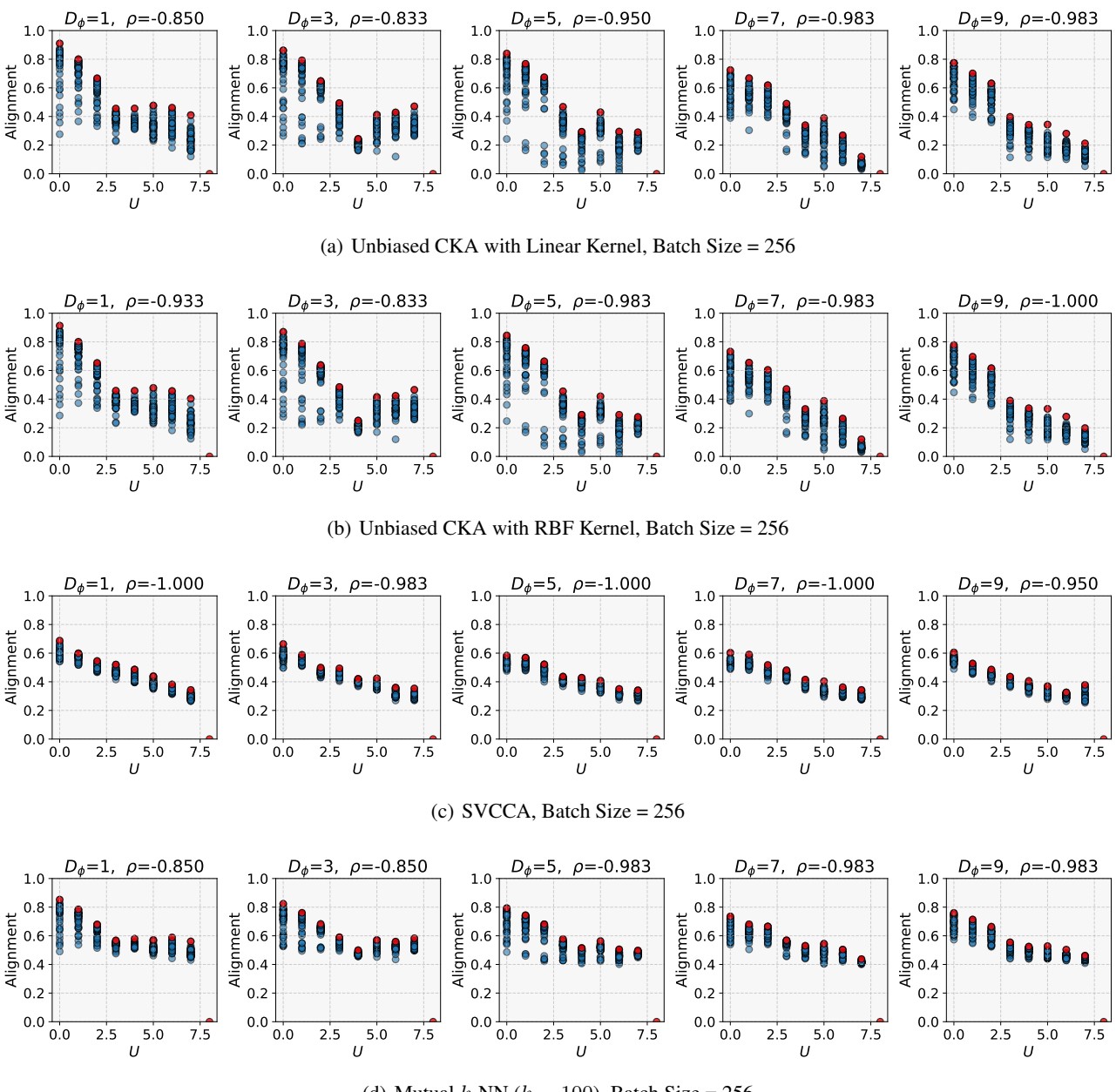

(a) Unbiased CKA with Linear Kernel, Batch Size = 256

(b) Unbiased CKA with RBF Kernel, Batch Size = 256

(c) SVCCA, Batch Size = 256

(d) Mutual $k$-NN ($k = 100$), Batch Size = 256

Figure 11: **Alignment vs uniqueness with batch size = 256.** Spearman correlation coefficient $\rho$ is computed between the maximum alignment, shown in red, and the level of informational uniqueness $U$.

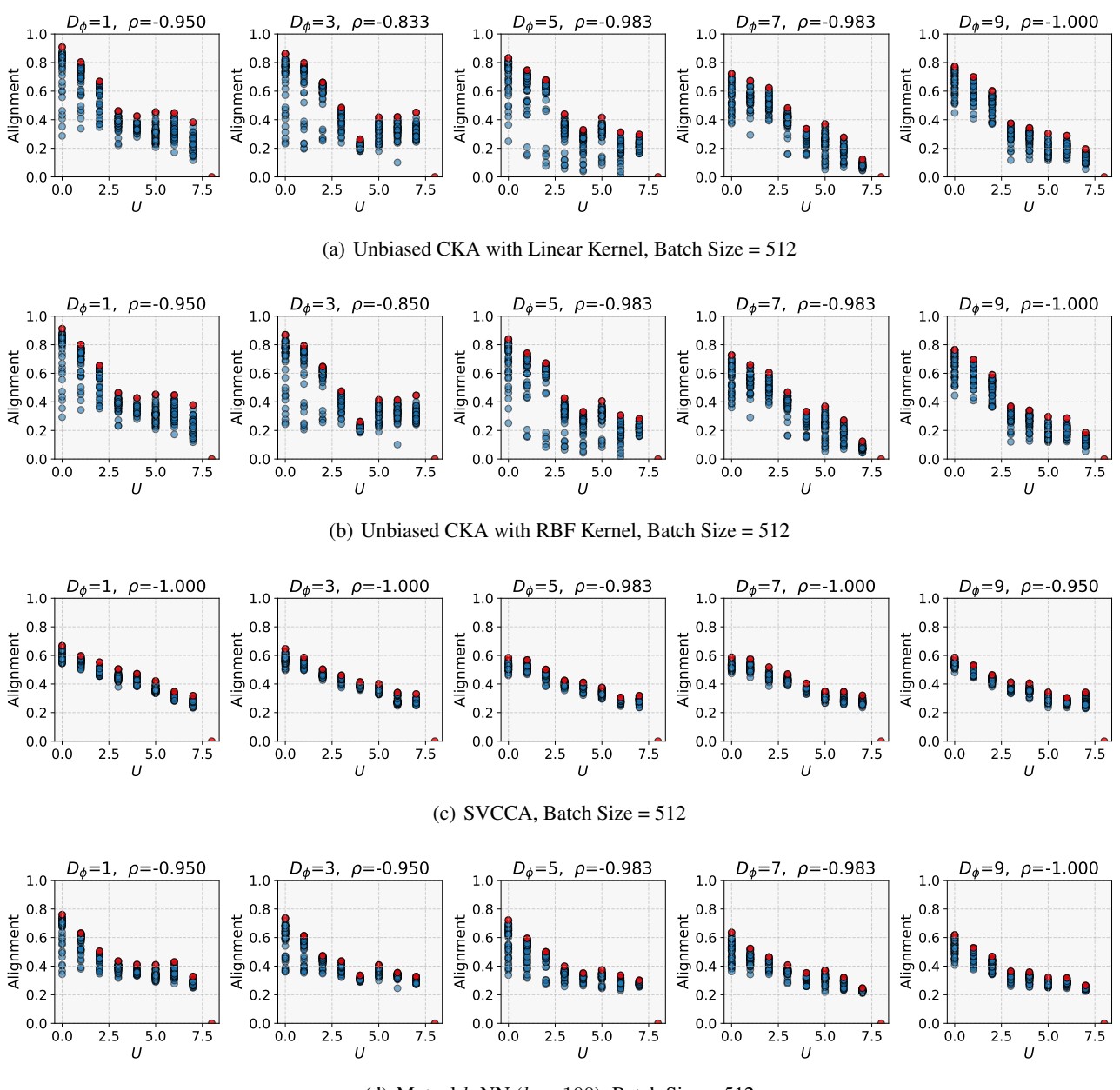

(a) Unbiased CKA with Linear Kernel, Batch Size = 512

(b) Unbiased CKA with RBF Kernel, Batch Size = 512

(c) SVCCA, Batch Size = 512

(d) Mutual $k$-NN ($k = 100$), Batch Size = 512

Figure 12: **Alignment vs uniqueness with batch size = 512.** Spearman correlation coefficient $\rho$ is computed between the maximum alignment, shown in red, and the level of informational uniqueness $U$.

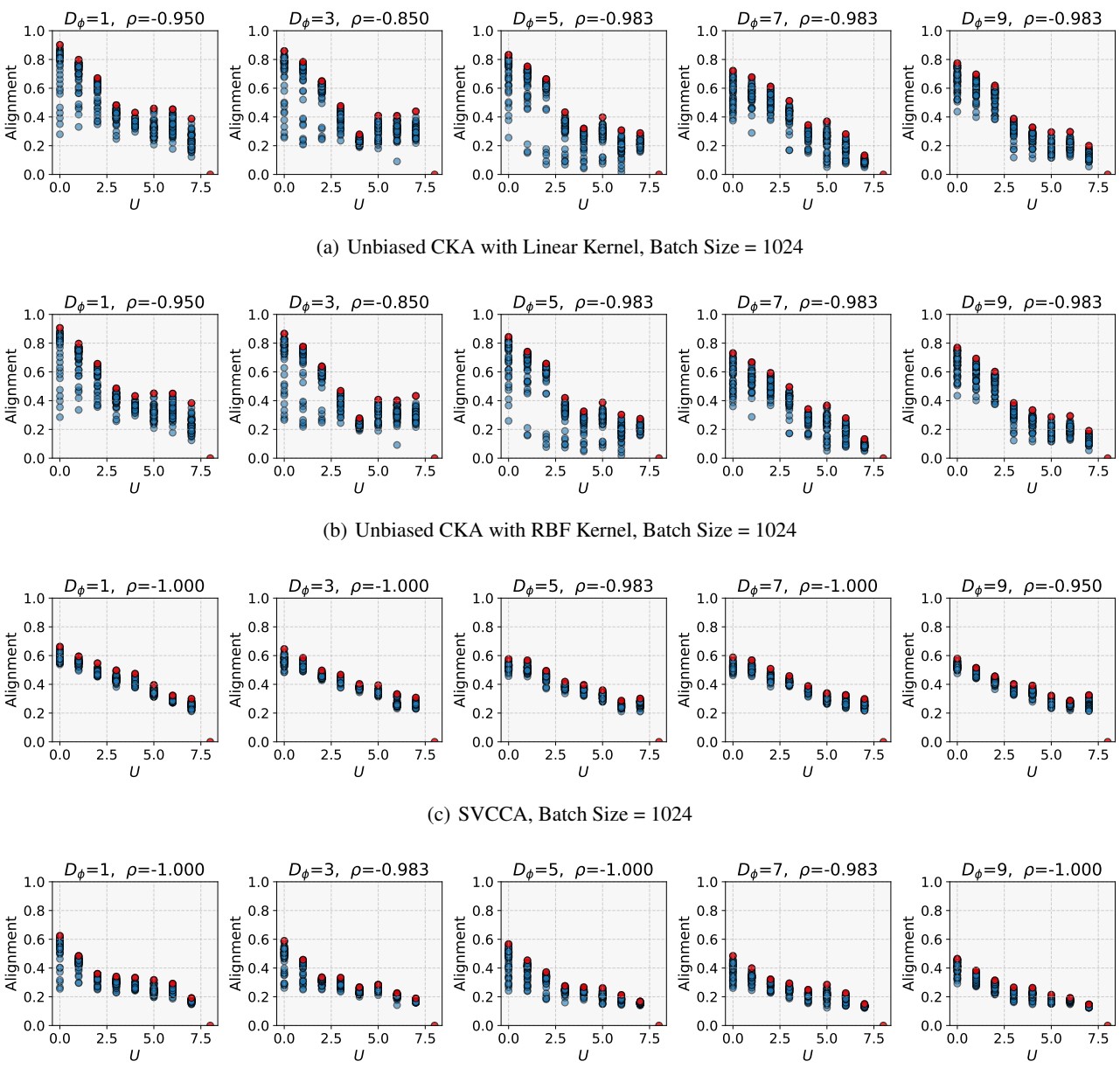

(a) Unbiased CKA with Linear Kernel, Batch Size = 1024

(b) Unbiased CKA with RBF Kernel, Batch Size = 1024

(c) SVCCA, Batch Size = 1024

(d) Mutual $k$-NN ($k = 100$), Batch Size = 1024

Figure 13: **Alignment vs uniqueness for various representation similarity metrics with batch size = 1024.** Spearman correlation coefficient $\rho$ is computed between the maximum alignment, shown in red, and the level of informational uniqueness $U$.

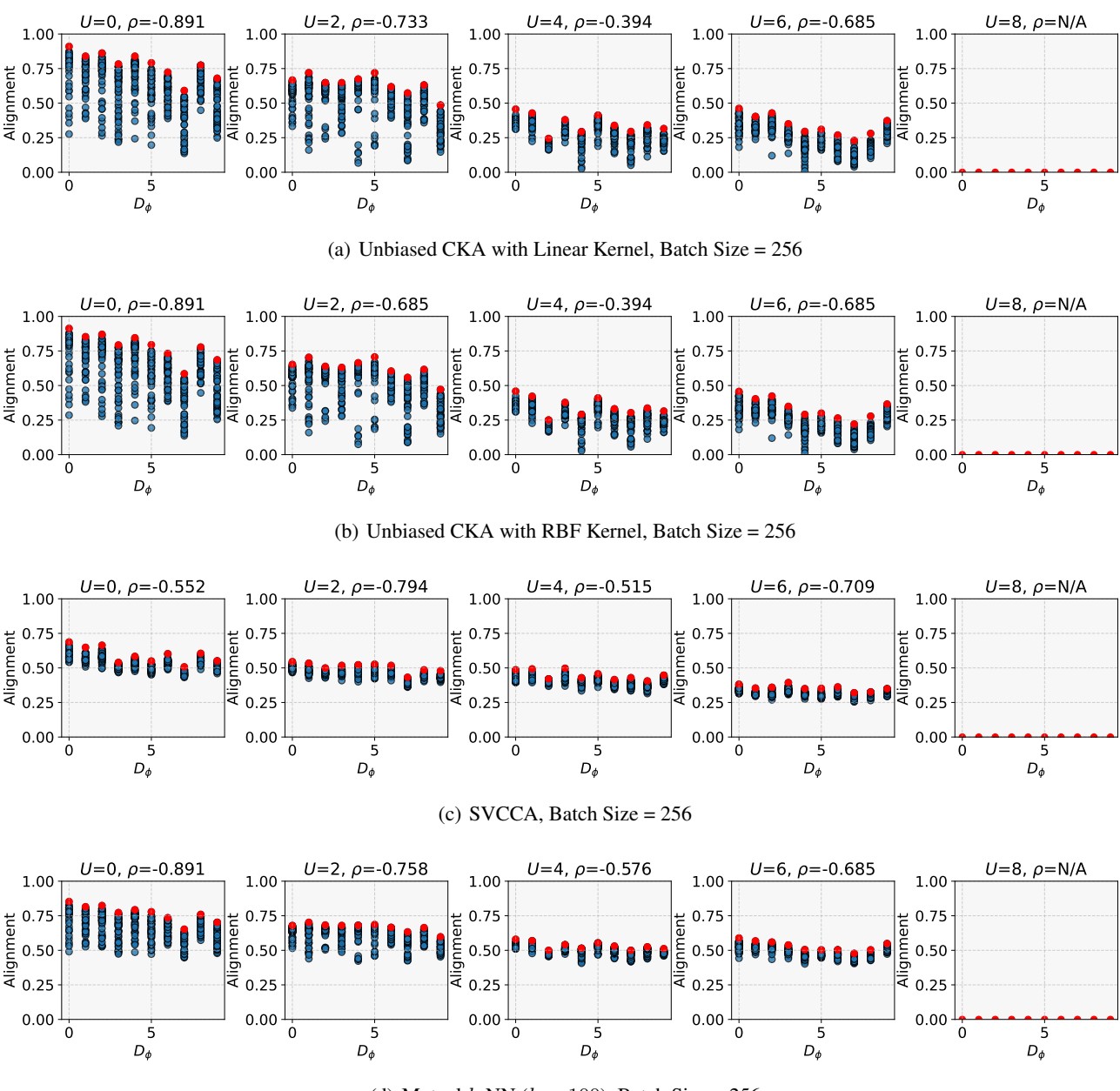

Figure 14: **Alignment vs heterogeneity for various representation similarity metrics with batch size = 256.** Spearman correlation coefficient $\rho$ is computed between the maximum alignment, shown in red, and heterogeneity. N/A denotes that $\rho$ is undefined as all alignment values are 0.

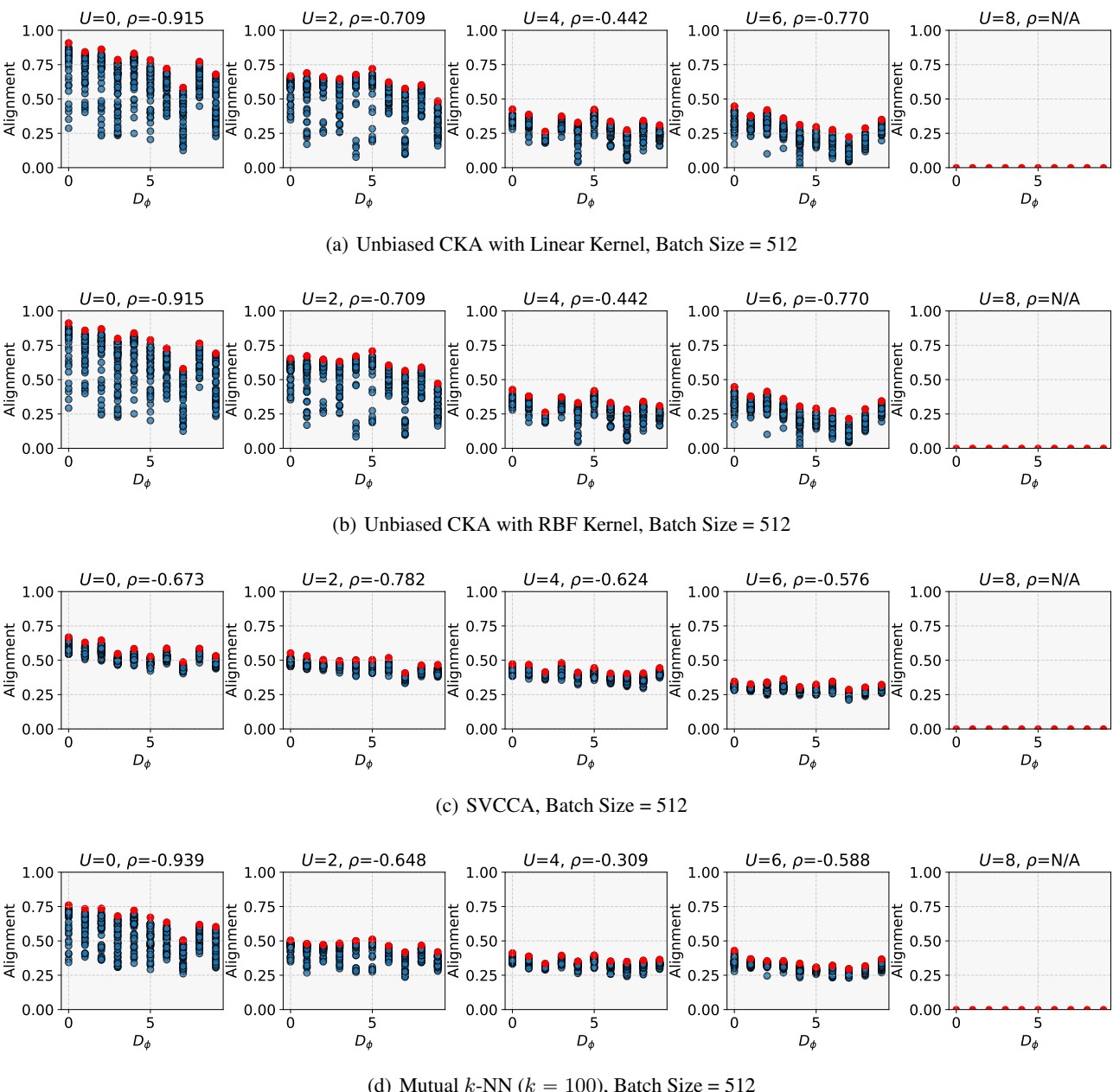

Figure 15: **Alignment vs heterogeneity with batch size = 512.** Spearman correlation coefficient $\rho$ is computed between the maximum alignment, shown in red, and heterogeneity. N/A denotes that $\rho$ is undefined as all alignment values are 0.

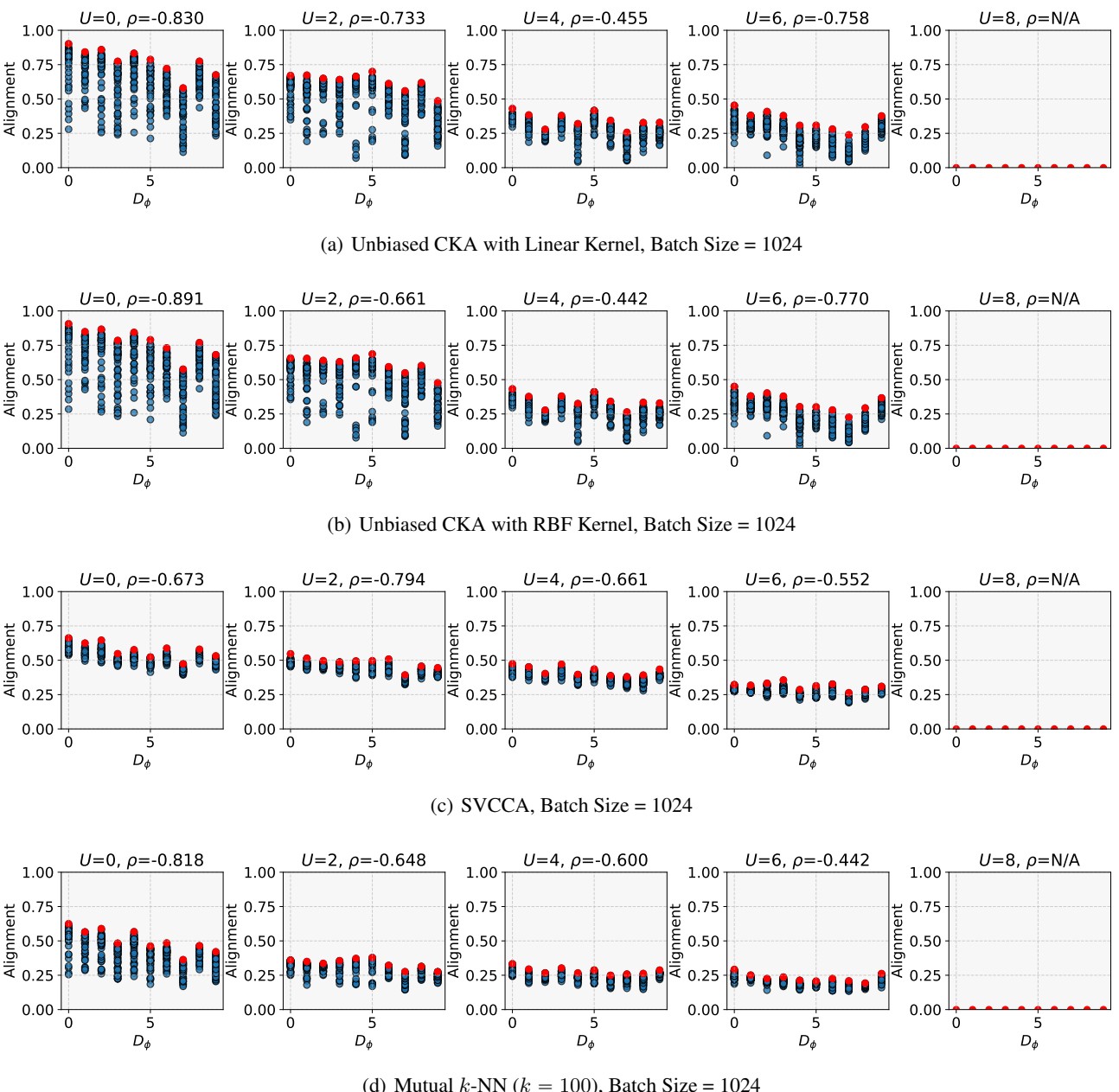

(a) Unbiased CKA with Linear Kernel, Batch Size = 1024

(b) Unbiased CKA with RBF Kernel, Batch Size = 1024

(c) SVCCA, Batch Size = 1024

(d) Mutual $k$-NN ($k = 100$), Batch Size = 1024

Figure 16: **Alignment vs heterogeneity with batch size = 1024.** Spearman correlation coefficient $\rho$ is computed between the maximum alignment, shown in red, and heterogeneity. N/A denotes that $\rho$ is undefined as all alignment values are 0.

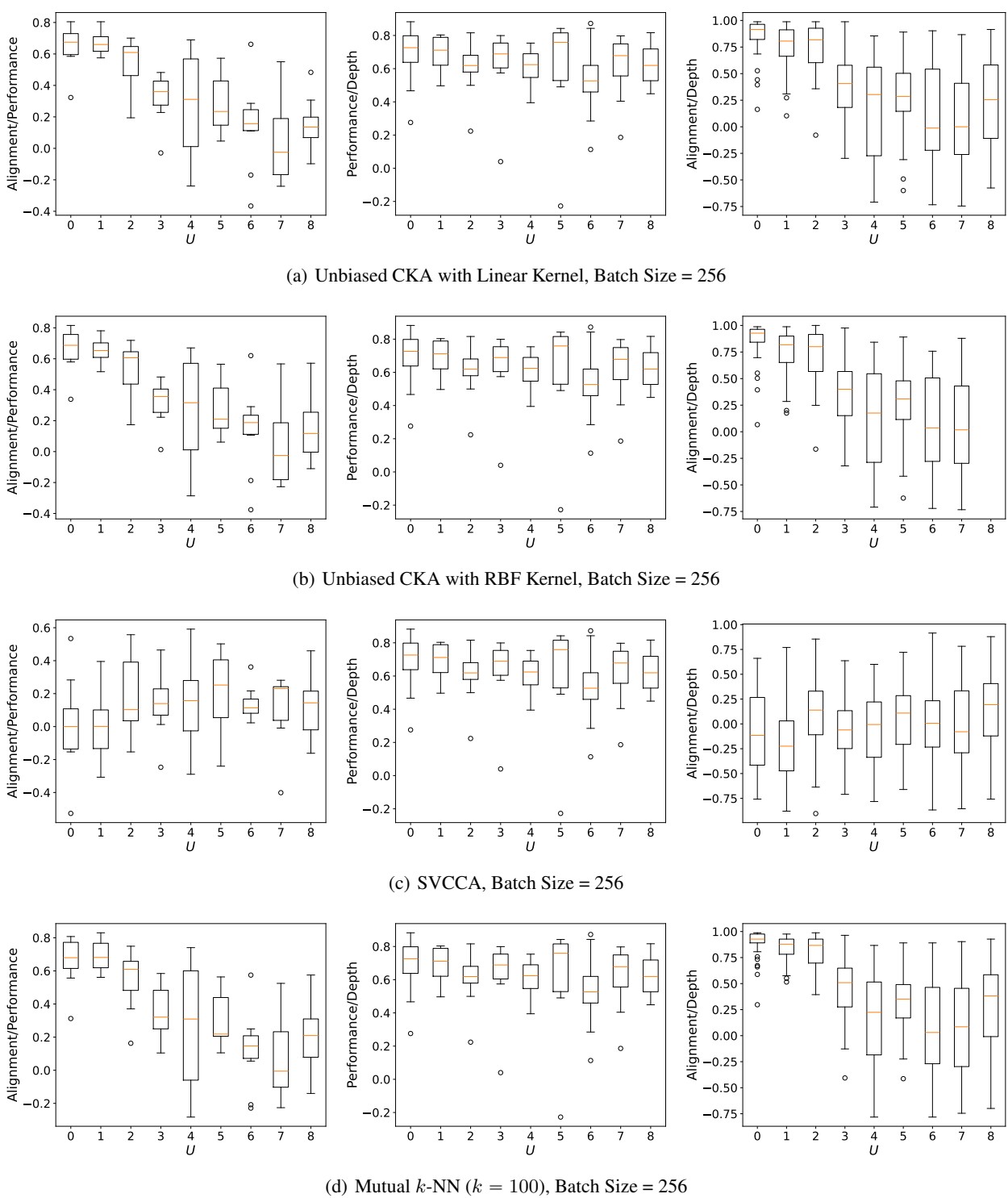

(a) Unbiased CKA with Linear Kernel, Batch Size = 256

(b) Unbiased CKA with RBF Kernel, Batch Size = 256

(c) SVCCA, Batch Size = 256

(d) Mutual $k$-NN ($k = 100$), Batch Size = 256

Figure 17: **Alignment, performance, and depth correlation plots across different synthetic depths and experiment seeds for various representation similarity metrics with batch size = 256.** In each plot, we show the spread of Spearman correlation coefficients $\rho$ for each level of uniqueness.

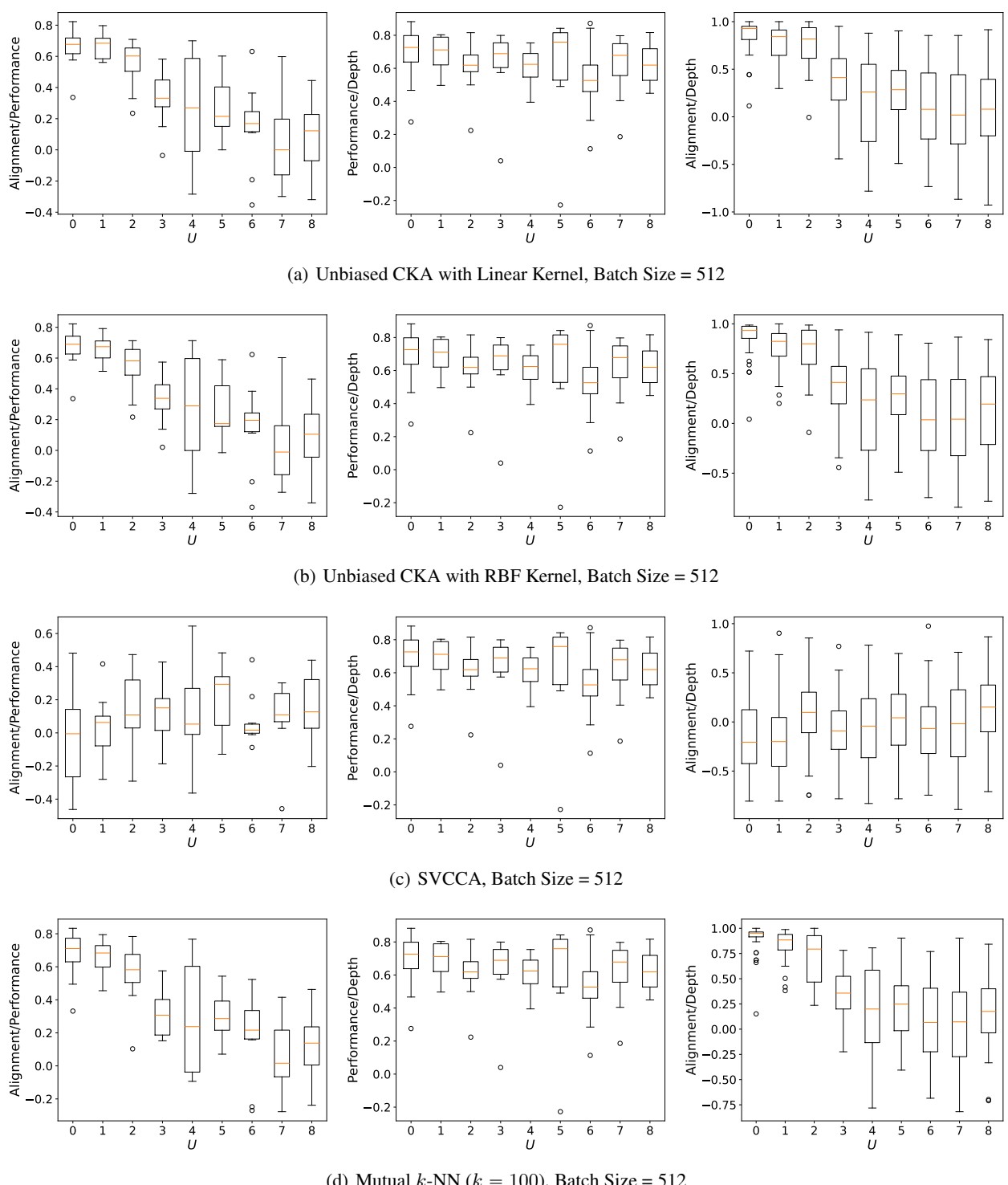

(a) Unbiased CKA with Linear Kernel, Batch Size = 512

(b) Unbiased CKA with RBF Kernel, Batch Size = 512

(c) SVCCA, Batch Size = 512

(d) Mutual $k$-NN ($k = 100$), Batch Size = 512

Figure 18: **Alignment, performance, and depth correlation plots across different synthetic depths and experiment seeds with batch size = 512.** In each plot, we show the spread of Spearman correlation coefficients $\rho$ for each level of uniqueness.

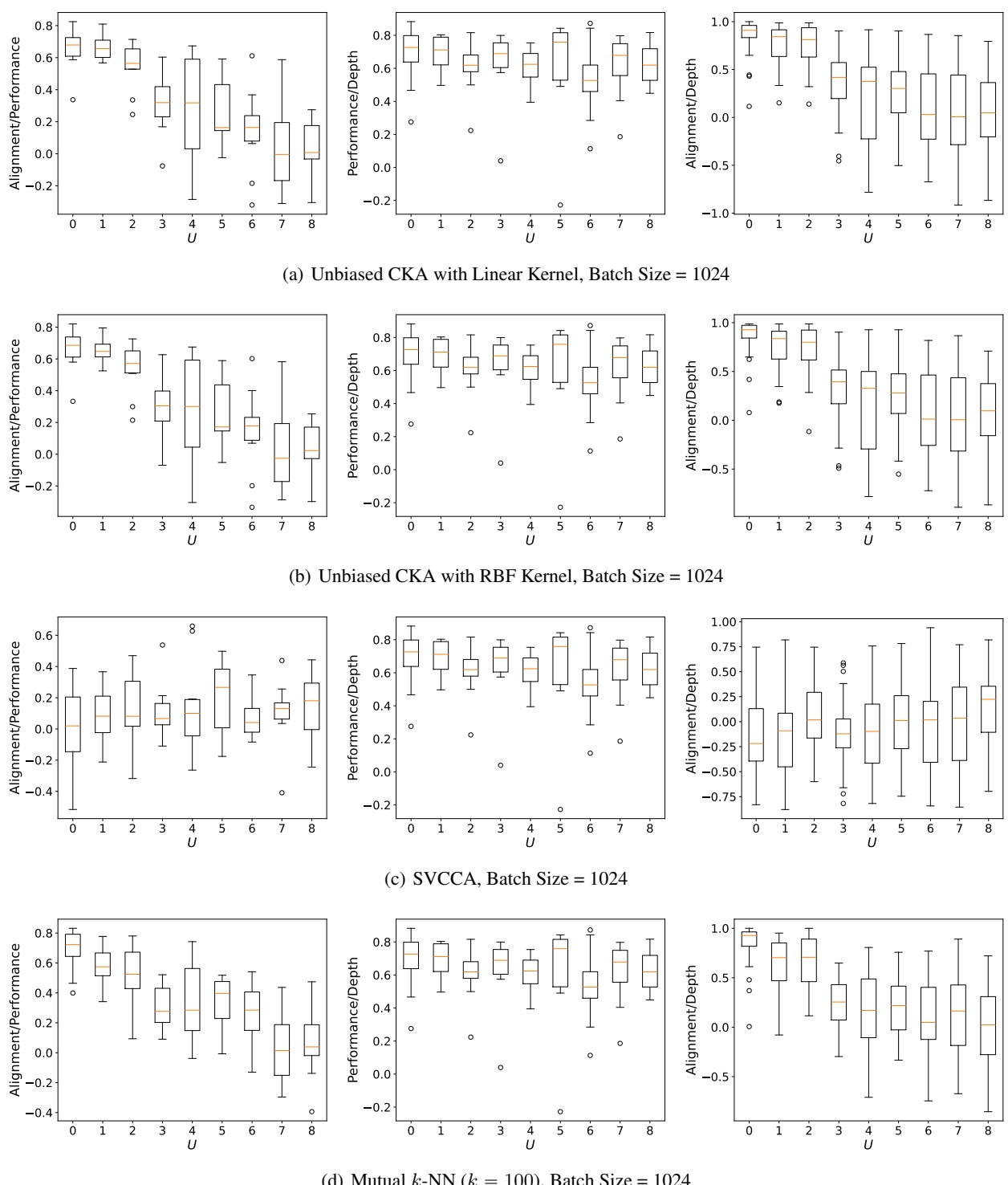

(a) Unbiased CKA with Linear Kernel, Batch Size = 1024

(b) Unbiased CKA with RBF Kernel, Batch Size = 1024

(c) SVCCA, Batch Size = 1024

(d) Mutual $k$-NN ($k = 100$), Batch Size = 1024

Figure 19: **Alignment, performance, and depth correlation plots across different synthetic depths and experiment seeds with batch size = 1024.** In each plot, we show the spread of Spearman correlation coefficients $\rho$ for each level of uniqueness.

## D.2. Synthetic Data Results with Different $E_1$ depths

In Figures 20 and 21, we provide additional experiment results showing that our results are not significantly changed when we increase the depth of $E_1$ to 2 and 3. Because $E_1$ is trained on the untransformed modality, $E_1$ will remain relatively easy to optimize even as the depth increases.

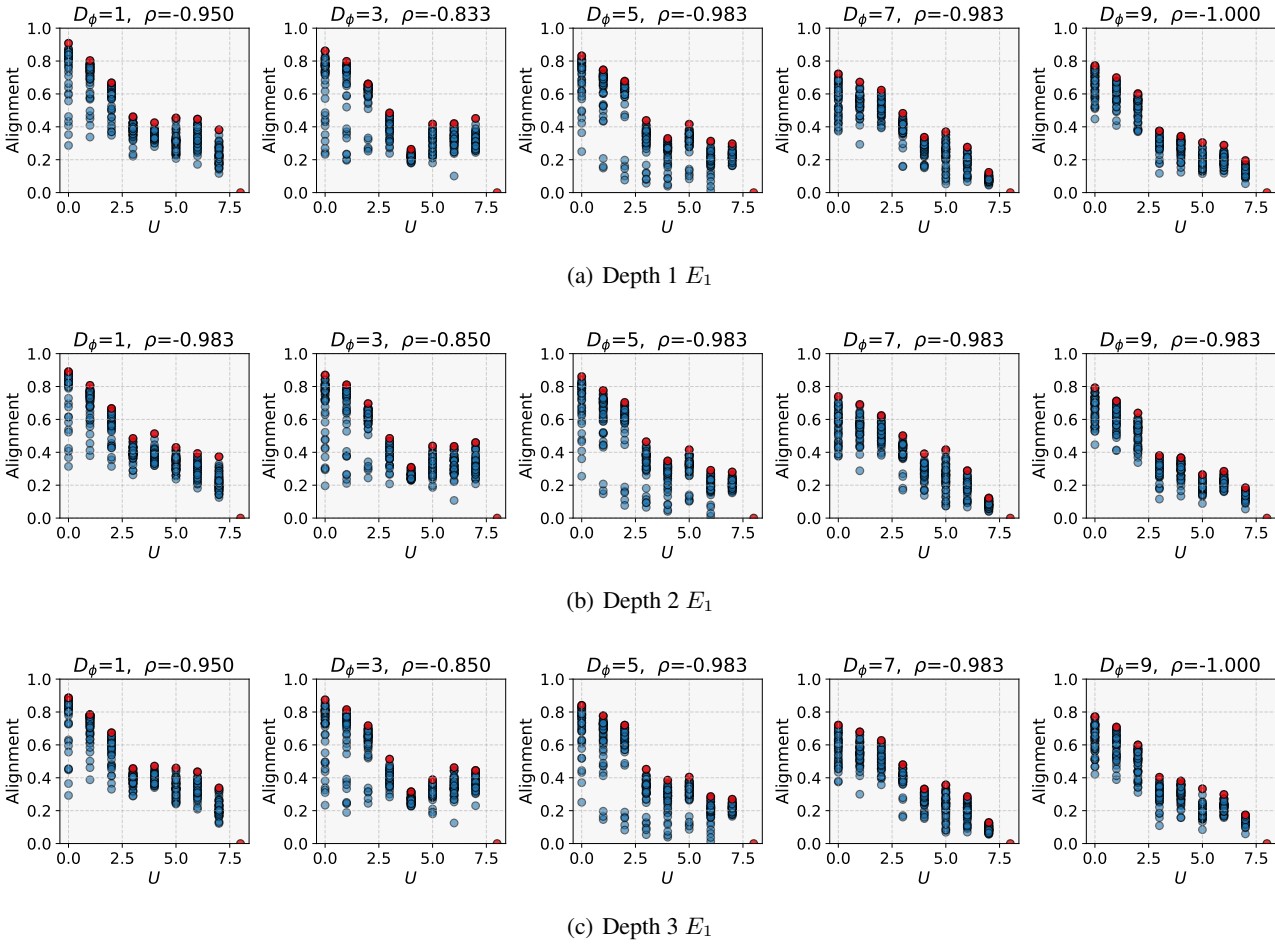

(a) Depth 1 $E_1$

(b) Depth 2 $E_1$

(c) Depth 3 $E_1$

Figure 20: **Alignment vs uniqueness for various depths of $E_1$.** The distribution of alignment scores for various depths of $E_1$ are nearly identical, as a single level neural network is sufficient to model the untransformed modality.

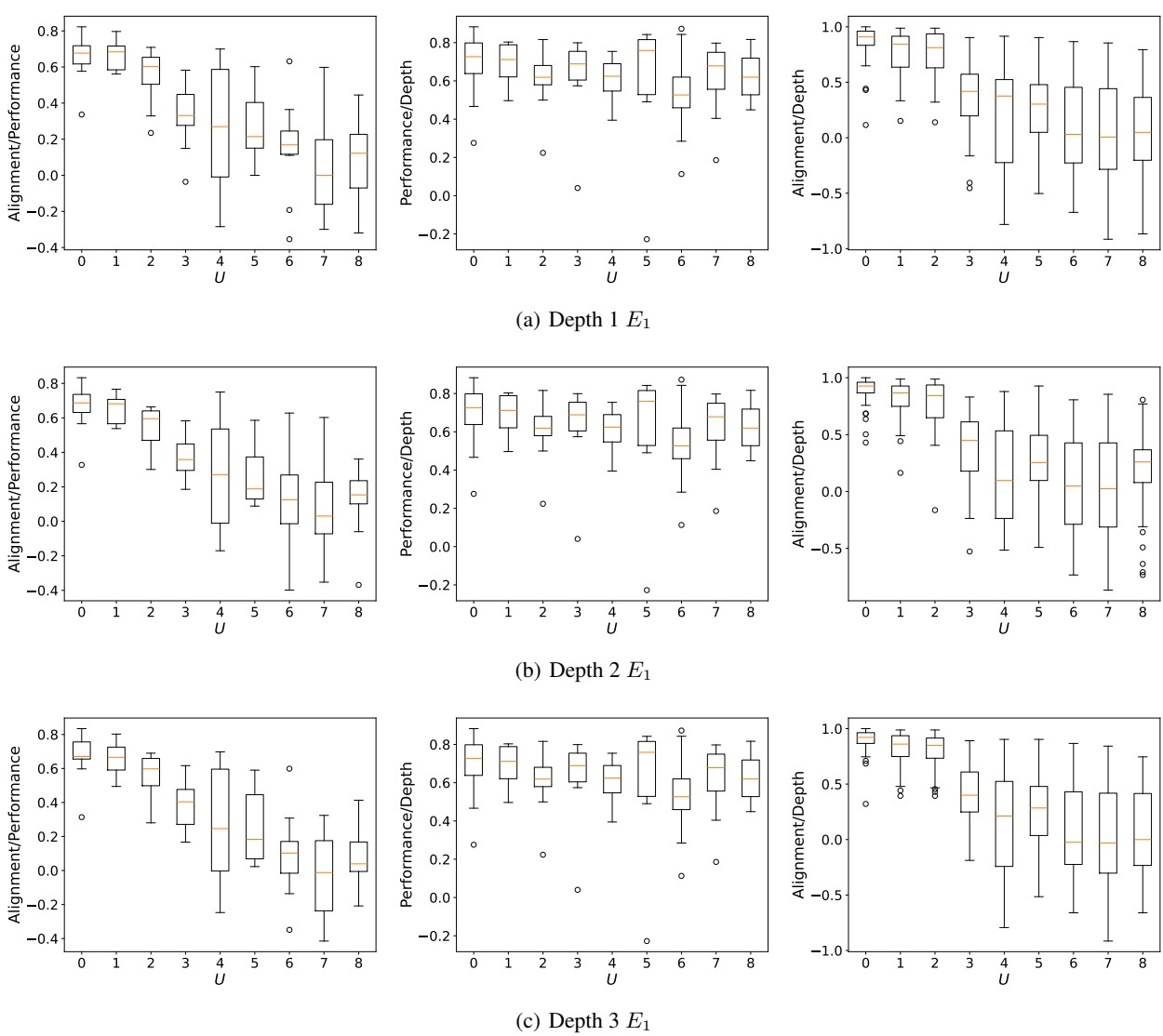

(a) Depth 1 $E_1$

(b) Depth 2 $E_1$

(c) Depth 3 $E_1$

Figure 21: **Alignment, performance, and depth correlation plots across different synthetic depths and experiment seeds for various depths of** $E_1$ In each plot, we show the spread of Spearman correlation coefficients $\rho$ for each level of uniqueness.

## D.3. Randomly Initialized Neural Networks Alignment

In Figure 22, we plot the alignment of randomly initialized neural networks. The alignment is constant for all levels of uniqueness, except for when the dataset is fully unique. In Figure 23, we show that for randomly initialized neural networks, alignment, performance, and depth do not correlate with each other.

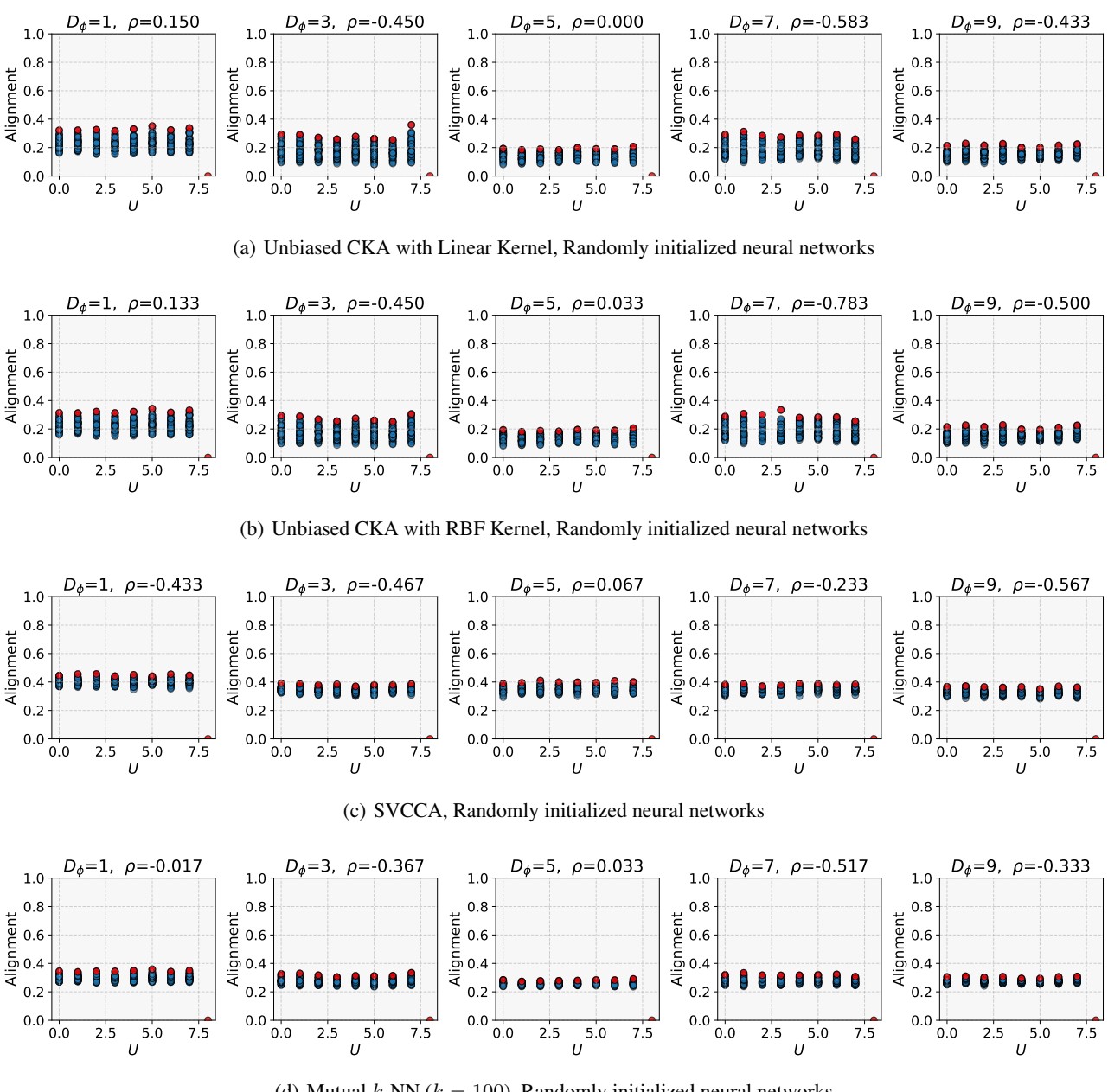

Figure 22: **Alignment vs uniqueness with randomly initialized neural networks.** Spearman correlation coefficient $\rho$ is computed between the maximum alignment, shown in red, and the level of informational uniqueness $U$.

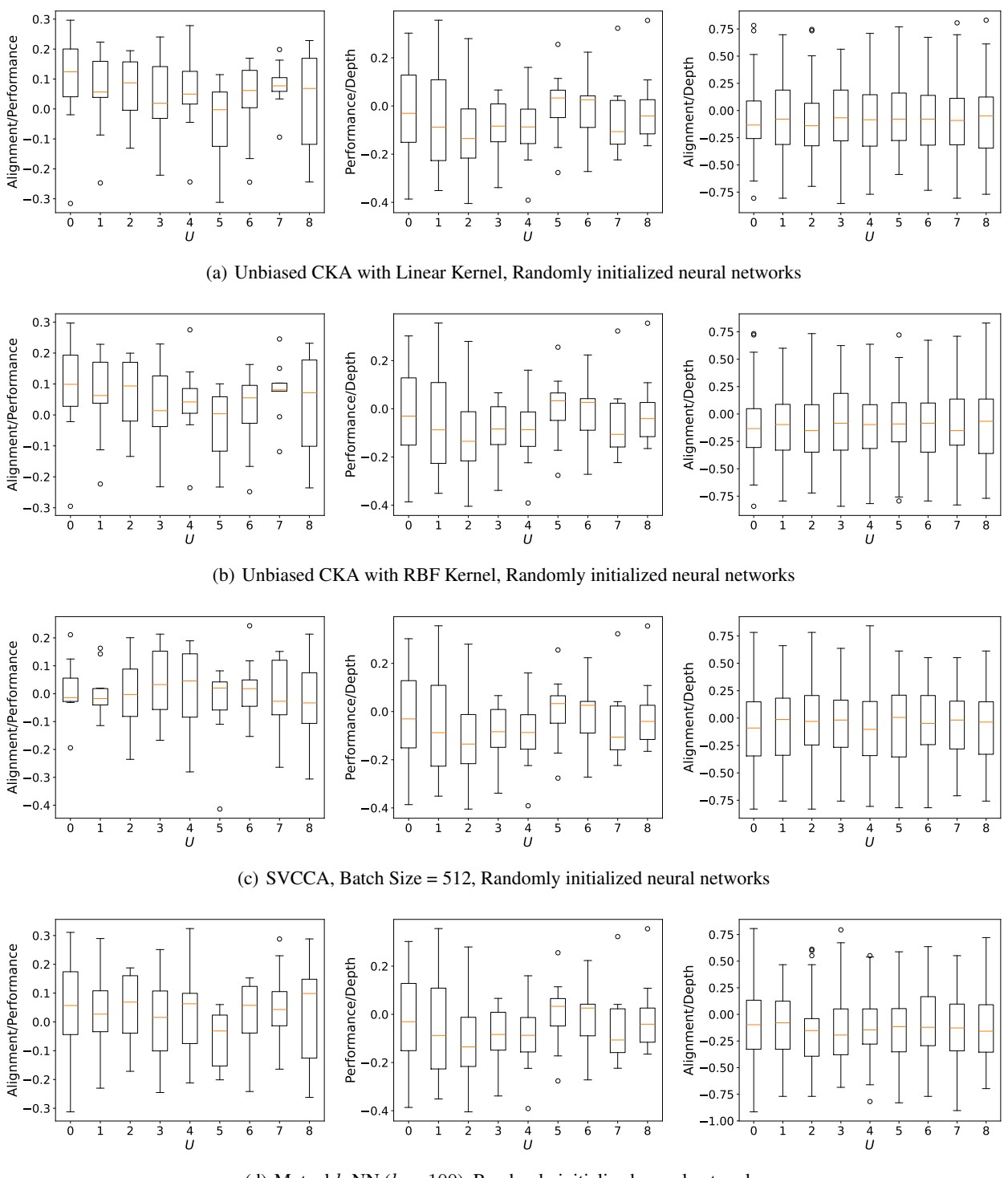

(a) Unbiased CKA with Linear Kernel, Randomly initialized neural networks

(b) Unbiased CKA with RBF Kernel, Randomly initialized neural networks

(c) SVCCA, Batch Size = 512, Randomly initialized neural networks

(d) Mutual $k$-NN ($k = 100$), Randomly initialized neural networks

Figure 23: **Alignment, performance, and depth correlation plots across different synthetic depths with randomly initialized neural networks.** In each plot, we show the spread of Spearman correlation coefficients $\rho$ for each level of uniqueness.

## D.4. Synthetic Data Alignment-Performance Results

In Figures 24, 25, and 26, we plot the relation between alignment and performance for individual synthetic datasets, which show that as uniqueness increases, alignment is no longer an indicator of performance.

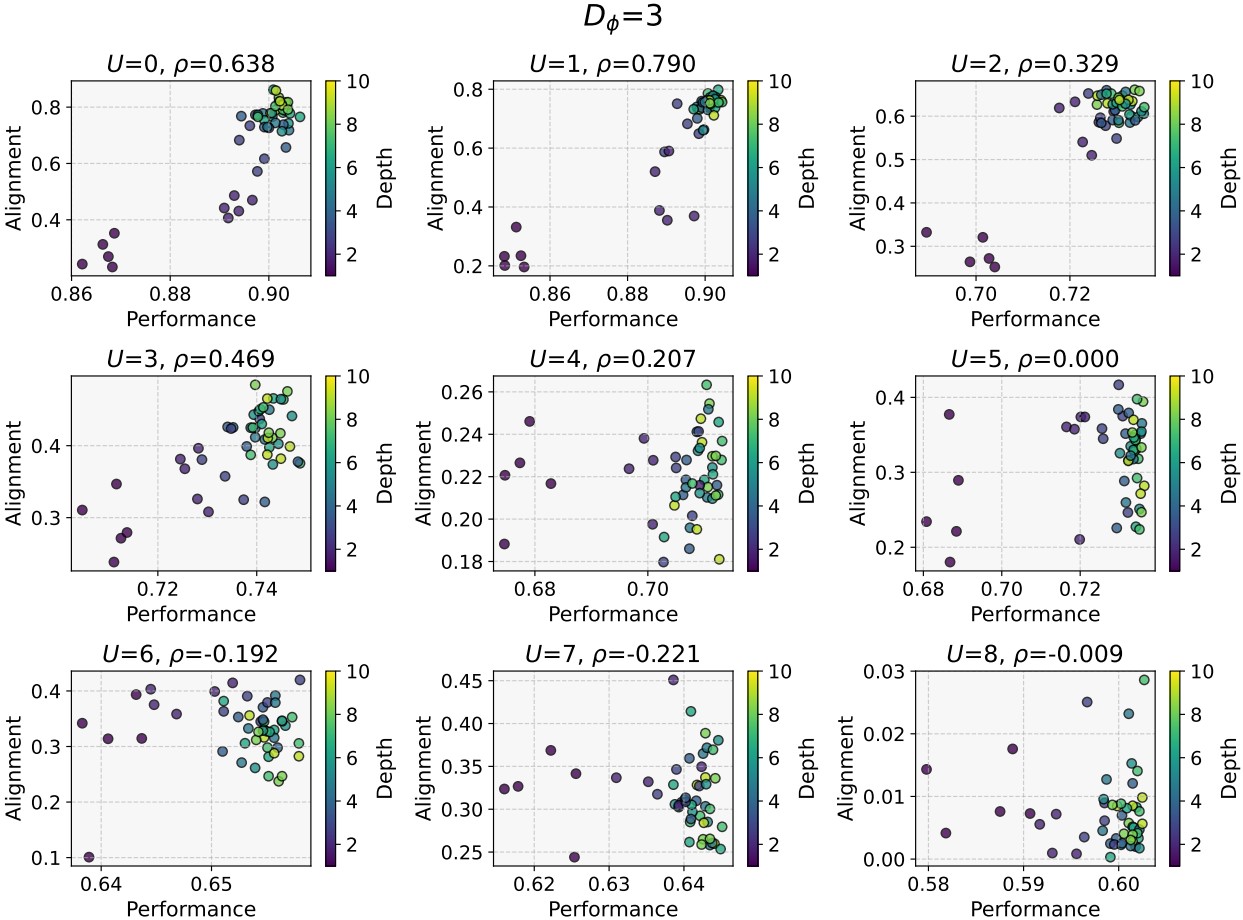

Figure 24: **Alignment vs Performance for** $D_\phi = 3$. The alignment-performance trend is shown across different levels of uniqueness, with the Pearson's correlation coefficient $r$ reported for each plot.

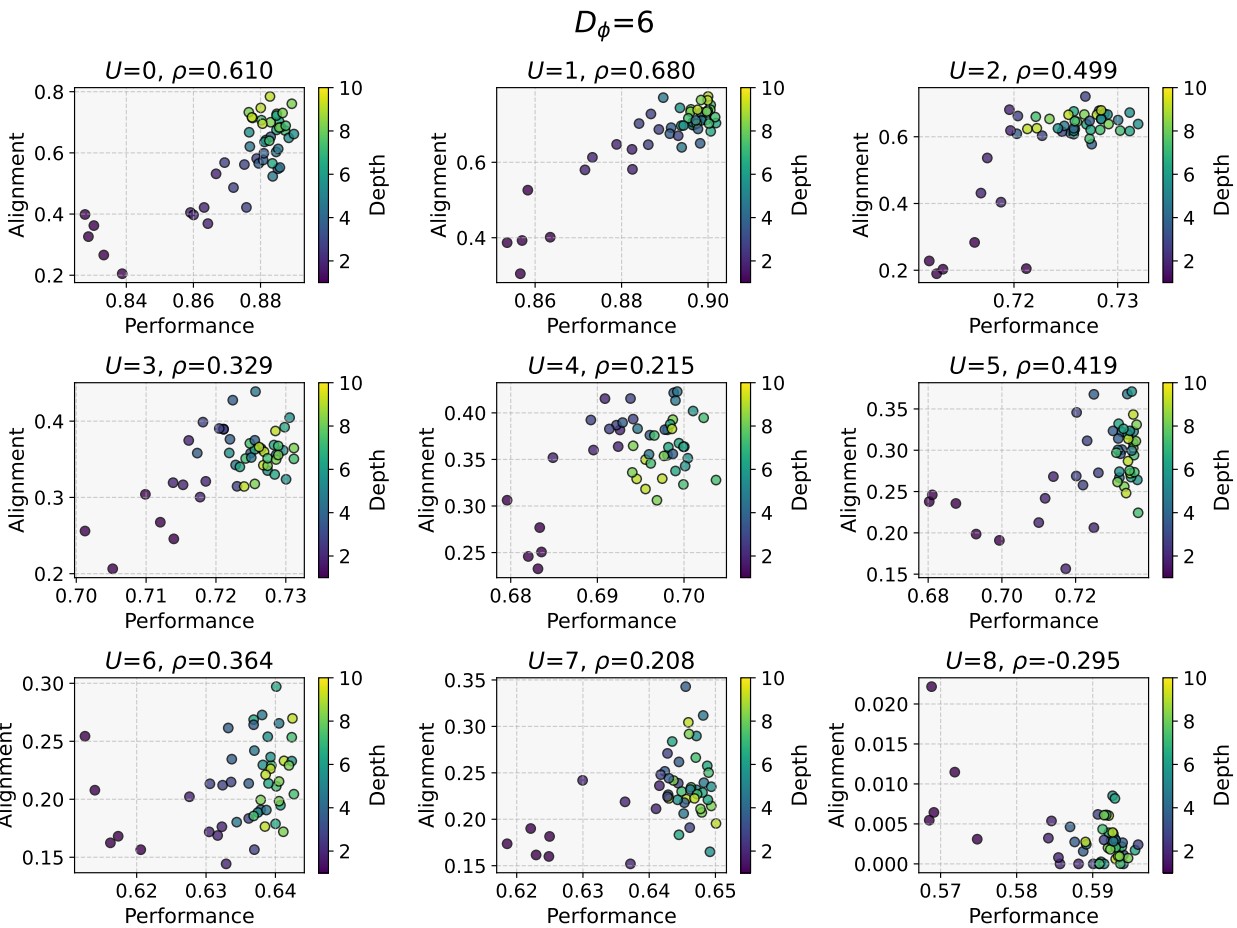

Figure 25: **Alignment vs Performance for** $D_\phi = 6$. The alignment-performance trend is shown across different levels of uniqueness, with the Pearson's correlation coefficient $r$ reported for each plot.

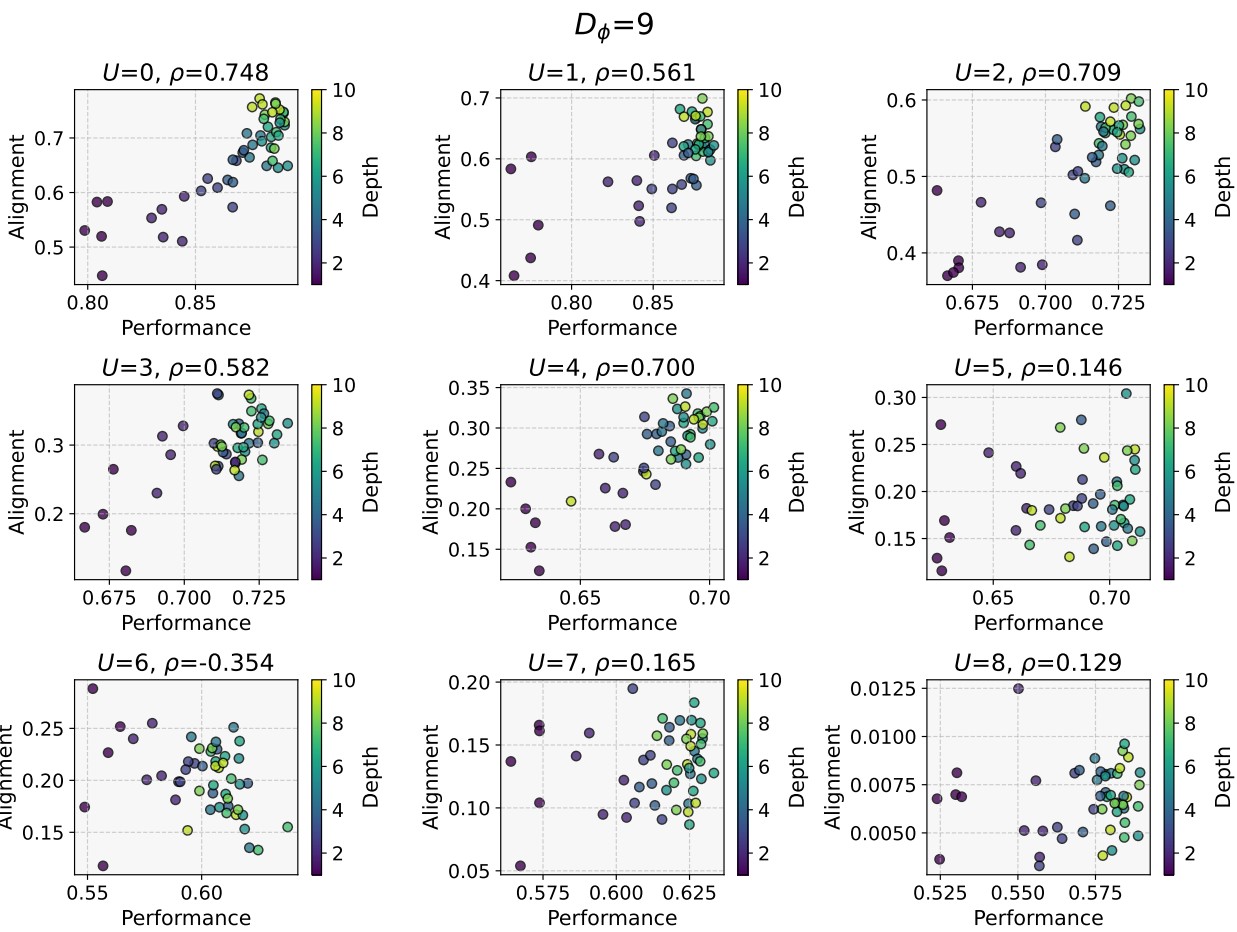

Figure 26: **Alignment vs Performance for** $D_\phi = 9$. The alignment-performance trend is shown across different levels of uniqueness, with the Pearson's correlation coefficient $r$ reported for each plot.

## D.5. Vision-Language Alignment vs Unique

In Figure 27, we plot the relation between vision-language alignment and uniqueness, which shows that the maximum alignment decreases with uniqueness.

## D.6. Vision-Language Alignment vs Performance

In Figure 28, we plot the relation between vision-language alignment and performance for various vision and language models.

## D.7. Alignment-performance relation on MM-IMDb

MM-IMDb (Arevalo et al., 2017) is a dataset for classifying movie genres from movie posters and text description of the movie plot, where there are 23 classes. As such, we consider 23 binary classification tasks. We compute cross-modal alignment between the same vision models and language models as Huh et al. (2024) using a subset of 1024 points. To obtain classification performance for each movie genre, we train linear classifiers using the last layer hidden representation of the language models. We compute the linear fit to alignment-performance scores for each downstream classification task. Intuitively, as the text describes the plot of the movie, we expect that the text modality provides many degrees of unique information compared to the image. However, not all of the additional information provided by the text would be useful to the given classification task, and thus the relation between alignment and performance would vary for different genres. Our analysis in Figure 29 reveals that the linear fit slopes vary depending on the movie genre. Larger linear fit slopes to alignment-performance scores suggest that aligning modalities is more helpful.

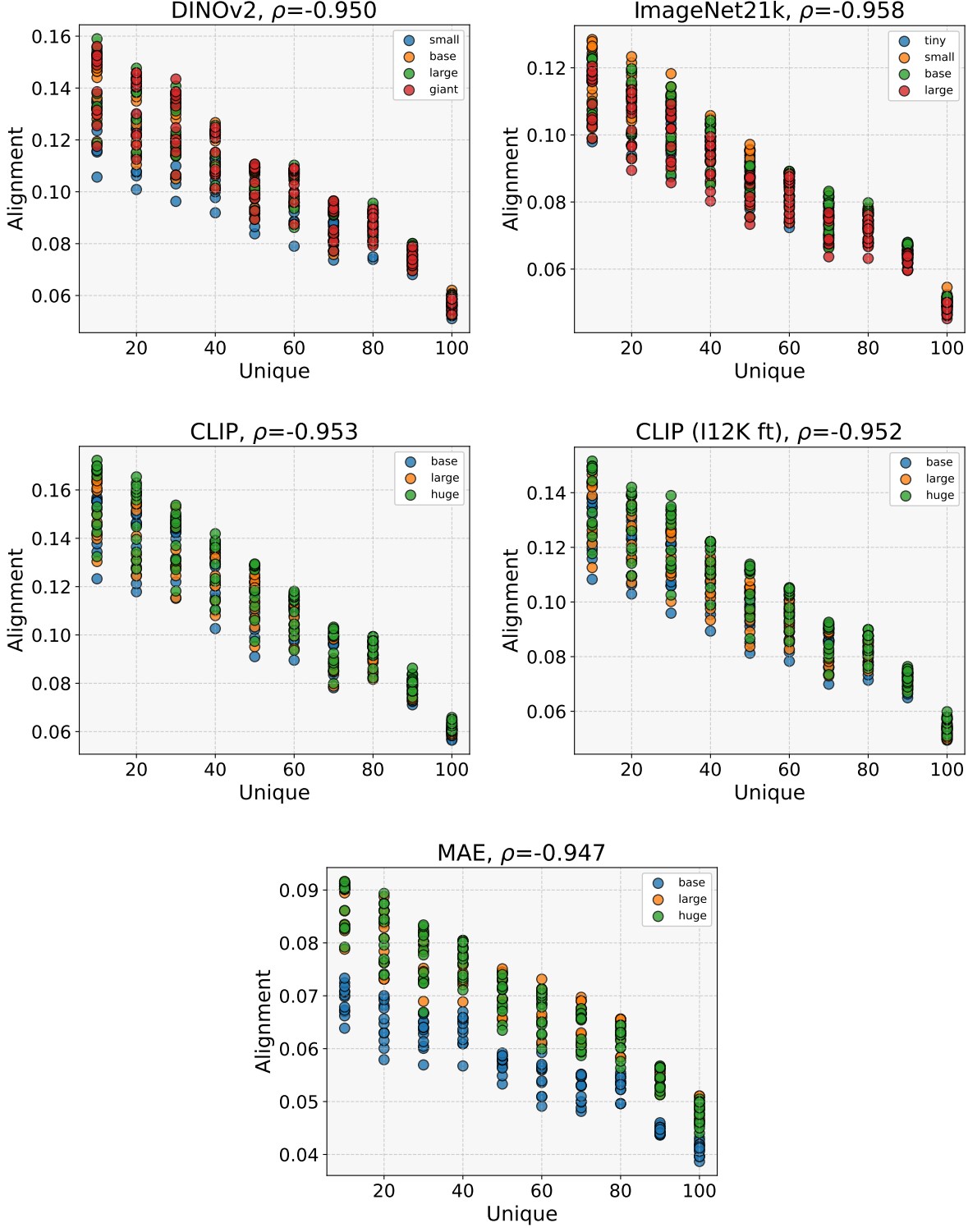

Figure 27: **Vision-Language Alignment vs Uniqueness.** The alignment is computed between various vision models and large language models. We compute the Spearman correlation coefficient $\rho$ between the maximum alignment and uniqueness.

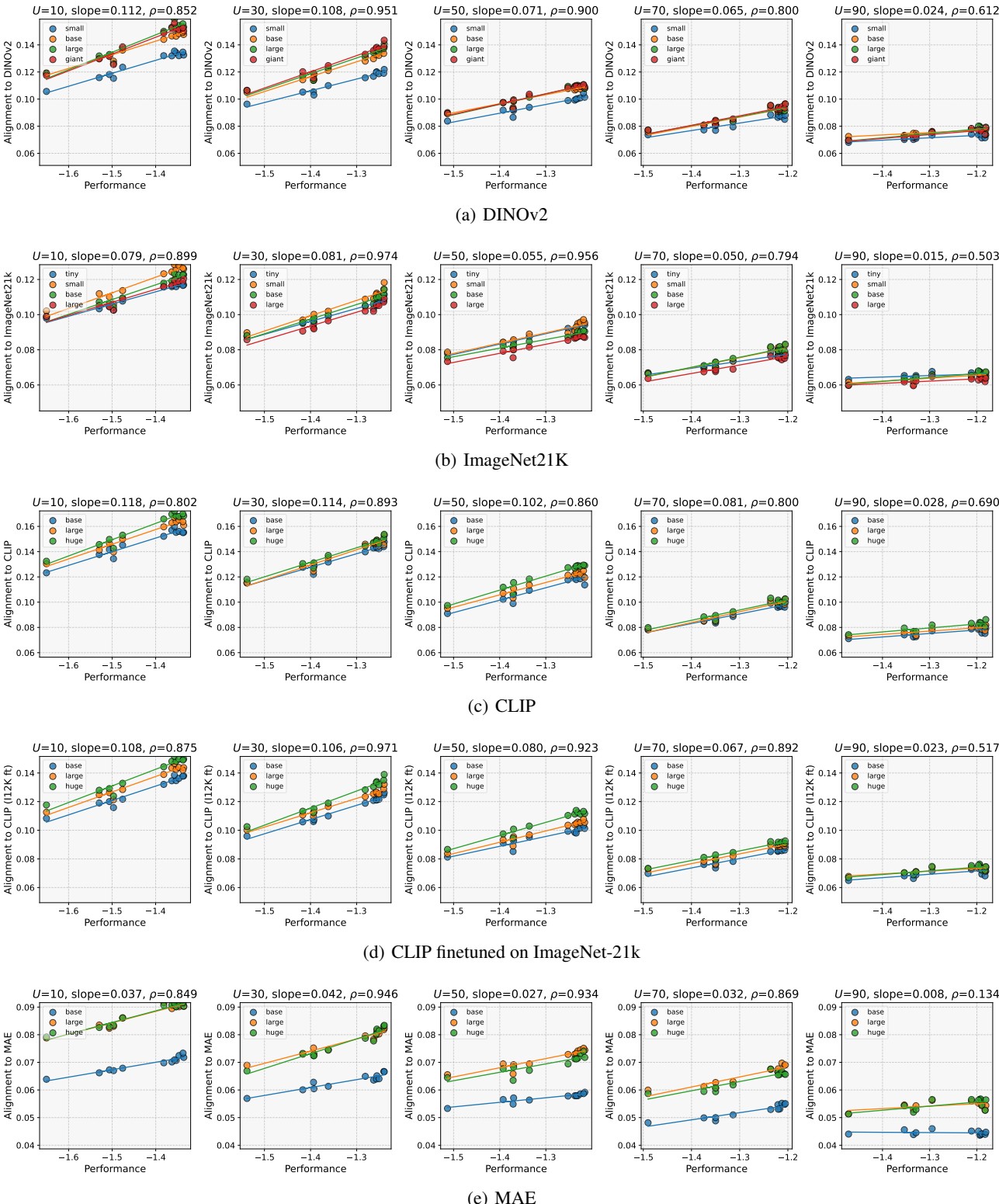

Figure 28: **Vision-Language Alignment vs Performance.** We plot the vision-language alignment using various vision models with respect to language model performance, measured using `bits-per-byte-loss` and show individual best fit lines for each size of vision model as well as the average Spearman correlation coefficient $\rho$. As $U$ increases, the relation between alignment and performance weakens.

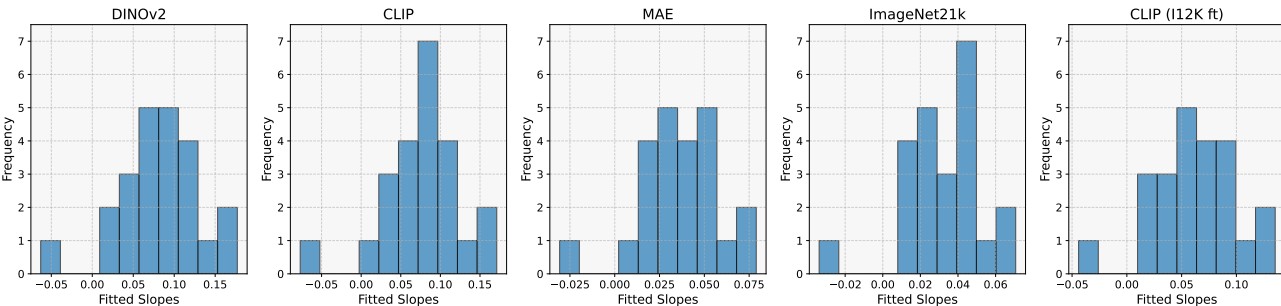

Figure 29: **Relationship between alignment and performances across MM-IMDb classification tasks.** Using the same set of language and vision models as Huh et al. (2024), we evaluate cross-modal alignment on MM-IMDb (Arevalo et al., 2017), a dataset for movie genre prediction, where we consider two modalities: images of the movie poster and text of the plot descriptions. The task is multi-label classification, and we consider each cateogry as a separate binary classification task. To measure performance for each language model, we train a linear classification layer on the last layer hidden representations. We plot the slope of the linear fit to the alignment-performance scores across categories, which varies due to different levels of information content required for each downstream classification task.

