# OpenReview forum: "Understanding the Emergence of Multimodal Representation Alignment"
_ICML.cc/2025/Conference — ICML 2025 poster_

### Official Review · Reviewer_nSuf · 2025-02-25

**Overall Recommendation:** 4

**Summary:**

This paper aims to understand the properties under which alignment emerges in multi-modal models. Specifically, they studied the influence of the data similarity (heterogeneity, i.e., how similar are two modalities) and uniqueness/redundancy of information (a.k.a. information imbalance) between the modalities on alignment (Figs. 2, 4-6, 10, 14-17). Alignment is measured via Huh et al’s [1] KNN-based center kernel alignment variant (also see Appendix B). Further, they studied how alignment correlates with performance (Figs. 7-9, 11-13, 18-22, Tab. 1). To answer these questions, they designed a synthetic dataset (Fig. 3) to control for uniqueness and heterogeneity. They corroborate the synthetic results with experiments using the Wikipedia caption dataset [5] and MultiBench [2].

## update after rebuttal

Please see my rebuttal comment below.

---

## References

[1] Huh, Minyoung, et al. "The platonic representation hypothesis." arXiv preprint arXiv:2405.07987 (2024).

[2] Liang, Paul Pu, et al. "Multibench: Multiscale benchmarks for multimodal representation learning." Advances in neural information processing systems 2021.DB1 (2021)

[3] Liang, Victor Weixin, et al. "Mind the gap: Understanding the modality gap in multi-modal contrastive representation learning." Advances in Neural Information Processing Systems 35 (2022): 17612-17625.

[4] Schrodi, Simon, et al. "Two effects, one trigger: on the modality gap, object bias, and information imbalance in contrastive vision-language representation learning." arXiv preprint arXiv:2404.07983 (2024).

[5] Srinivasan, Krishna, et al. "Wit: Wikipedia-based image text dataset for multimodal multilingual machine learning." Proceedings of the 44th international ACM SIGIR conference on research and development in information retrieval. 2021.

**Claims And Evidence:**

* Maximum achievable alignment is controlled by uniqueness of the input modalities. This is well-supported by the experimental evidence in Figs. 2, 4-6, 14-17 on synthetic as well as real data. The results for heterogeneity (Figs. 6, 10) are less clear.

* Performance is not directly correlated to alignment (Sec. 5). This is again supported by Figs. 7, 8. Again, the results for heterogeneity seem less clear (Fig. 9).

**Essential References Not Discussed:**

* Schrodi et al [4] showed that information imbalance causes the modality gap and object bias. Information (im)balance (called information redundancy/uniqueness in this work) is also the data property studied in this work (besides heterogeneity). Particularly, they hypothesized that less shared information worsens alignment that leads to the modality gap and object bias. Further, some findings and experiments share a resemblance. Thus, it’d be good to discuss the similarities and differences to Schrodi et al. in future versions.

**Experimental Designs Or Analyses:**

* The synthetic dataset is well-designed and motivated to cleanly study the effect of uniqueness and heterogeneity (Fig. 3).

* Correlation is measured by the Pearson correlation coefficient. However, rank-based correlation coefficients would be a bitter fit, like Spearman or Kendall’s $\tau$, since they don’t assume a relationship a priori (beyond that the order should matter).

**Methods And Evaluation Criteria:**

* The KNN-based variant of center kernel alignment based on Huh et al [1] is well-suited to evaluate alignment. However, other alignment measures would be appreciated since properly measuring alignment is challenging.

* The synthetic and real datasets, as well as chosen models are well-suited.

**Other Comments Or Suggestions:**

* It’d be good to make more explicit how alignment is measured between visual-only models like DINOv2 and the LLMs, as done by Huh et al [1].

**Other Strengths And Weaknesses:**

* S: the paper is well-written and clear.

**Questions For Authors:**

* In the synthetic data, is information always redundant or unique for each sample or can this vary per-sample? E.g., for one sample a factor is part of the data while for another sample it is not.

* Why does the first encoder in the synthetic setting only have a single layer?

* How are the models trained in the synthetic setting? I.e., what type of training method is used? CLIP loss? Captioning loss?

* How are correlations computed? Since there are many points per x-value across all plots, this should lower correlations, right?

* What is the effect of the number of task-relevant features? Currently, it is only set to 8. What happens when you set it to 256?

**Relation To Broader Scientific Literature:**

Huh et al [1] put forward the platonic representation hypothesis. This paper investigates key data properties (information balance and data heterogeneity) on the alignment. I’d like to note that work by Schrodi et al [4] also investigated information imbalance in the context of the modality gap and object bias for CLIP models (see below for more details). Findings and experiments seem related, though the scope is different. Thus, I conclude that this work is a valuable contribution on understanding how data shapes the models, in this case their representational alignment.

**Theoretical Claims:**

N/A

---

> ### Author Rebuttal · Authors · 2025-04-01
>
> We thank the reviewer and are glad that they find our experiments well-designed and motivated. Below we address the reviewer’s comments and questions.
>
> **Under “Methods And Evaluation Criteria:”**
> > The KNN-based variant … other alignment measures would be appreciated since properly measuring alignment is challenging.
>
> We report additional results with unbiased CKA with a RBF kernel, Mutual KNN [1], SVCCA [2] with three different sample sizes, and all metrics support our paper’s main claims. See response to reviewer B2rQ for more details.
>
> [1] Huh et al. "The platonic representation hypothesis." (2024).
> [2] Raghu et al. “SVCCA: Singular Vector Canonical Correlation Analysis for Deep Learning Dynamics and Interpretability” (2017).
>
> **Under “Experimental Designs Or Analyses:”**
> > Correlation is measured by the Pearson correlation coefficient. However, rank-based correlation coefficients would be a bitter fit, like Spearman or Kendall’s $\tau$, since they don’t assume a relationship a priori (beyond that the order should matter).
>
> We are happy to explore other correlation metrics. [Here](https://tinyurl.com/h3fv2ez3) are the results using Spearman correlation. We find that the Spearman correlation supports our main claims: it shows a strong negative trend between maximum alignment and uniqueness and that the relation between alignment and performance is weaker or negative when there is greater uniqueness. Also, we note the Pearson correlation might actually be a suitable choice given common observations that linear relations in latent representations tend to emerge after training (e.g., see https://arxiv.org/abs/2007.00810).
>
> **Under “Essential References Not Discussed:”**
> > Schrodi et al [4] showed that information imbalance causes the modality gap and object bias. …It’d be good to discuss the similarities and differences to Schrodi et al. in future versions.
>
> Thank you for bringing up this related work. We agree that [4] is related to our work in that it analyzes the effect of information imbalance on the representations learned through contrastive learning, whereas our work focuses on emerging alignment through increased model capacity. We will add a discussion of [4] in our updated paper.
>
> **Under “Other Comments Or Suggestions:”**
>
> > It’d be good to make more explicit how alignment is measured between visual-only models like DINOv2 and the LLMs, as done by Huh et al [1].
>
> The details of the alignment computation are in Appendix B as follows: “Following Huh et al. (2024), we use $k=10$ nearest neighbors over 1024 samples from the Wikipedia caption dataset. For the vision model, the class token of each layer is used, and for the language model, the embeddings of a given layer are average pooled to a single token. $l_2$ normalization is applied to the features and elements in the features that are above the 95-th percentile are truncated.”
>
> We’re happy to add any additional details that the reviewer thinks are missing.
>
> **Under “Questions For Authors:”**
>
> > In the synthetic data, is information always redundant or unique for each sample or can this vary per-sample? E.g., for one sample a factor is part of the data while for another sample it is not.
>
> The proportion of redundant to unique information is constant for all samples.
>
> > Why does the first encoder in the synthetic setting only have a single layer?
>
> We provide additional experiment results demonstrating that our results are unchanged when $E_1$ is a higher depth. [Here](https://tinyurl.com/4uwe4zkm), we change the depth of $E_1$ to 2 and 3 and find that the results are not significantly changed. We hypothesize that because $E_1$ is trained on the untransformed modality, $E_1$ will remain relatively easy to optimize even as the depth increases. We will include these results in our updated paper.
>
> > How are the models trained in the synthetic setting? I.e., what type of training method is used? CLIP loss? Captioning loss?
>
> In the synthetic setting, the ground truth labels are available, the models are trained in a supervised manner, with cross-entropy loss.
>
> > How are correlations computed? Since there are many points per x-value across all plots, this should lower correlations, right?
>
> In Fig. 4 and 5, the correlation is computed using only the maximum alignment rather than all points. We will clarify this in our updated paper.
>
> > What is the effect of the number of task-relevant features? Currently, it is only set to 8. What happens when you set it to 256?
>
> The total number of task-relevant features would not impact our results -- what matters is the proportion of redundant to unique features. If we had 256 task-relevant features, out of which 128 are shared, we would expect to see that the result is similar to $U=4$ in our setting. Our results on real-world data are on much higher dimensions -- for our experiments on Wikipedia Image Text and on MM-IMDb, we use LLMs with high dimensional latent spaces of 1024 or greater.

---

> > ### Comment · Reviewer_nSuf · 2025-04-02
> >
> > I thank the reviewer for their replies to others and my reviews. I tend to uphold my score of 4 despite the critiques brought up by other reviews. In particular, I think the chosen alignment metric is a justified choice and the added evaluations using other metrics provides sufficient evidence to support the authors’ claims.
> >
> > That said, I do have a follow-up question regarding “is information always redundant or unique” since the response has not addressed the core of my question: What happens if certain information is redundant for some samples but unique for others? For example, color might be redundant in some cases but unique in others. How would this variability of whether information is redundant or unique affect the findings?

---

> > > ### Author Response · Authors · 2025-04-08
> > >
> > > > That said, I do have a follow-up question regarding “is information always redundant or unique” since the response has not addressed the core of my question: What happens if certain information is redundant for some samples but unique for others? For example, color might be redundant in some cases but unique in others. How would this variability of whether information is redundant or unique affect the findings?
> > >
> > > We thank the reviewer for this insightful and intellectually stimulating question. To recapitulate, the reviewer inquires whether the notions of redundancy and uniqueness, as used in our work, should be regarded as *aggregate*  quantities—computed over the joint distribution of variables—or whether a *pointwise* (i.e., sample-specific) formulation might be more appropriate or feasible.
> > > In our study, we adopt the definitions of redundancy, uniqueness, and synergy as formalized in the Partial Information Decomposition (PID) framework [1]. These definitions are intrinsically grounded in mutual information, which is, by construction, an expectation over the joint distribution of the relevant random variables. That is, mutual information quantifies average statistical dependence and does not inherently attribute information values to individual data points or observations. Consequently, the redundancy, uniqueness, and synergy measures derived from mutual information are likewise aggregate in nature: they describe global statistical properties of the system rather than localized or instance-specific contributions.
> > >
> > > We concur with the reviewer that it is conceptually plausible—and potentially of practical significance—to consider localized (e.g., pointwise or groupwise) versions of information-theoretic measures. In particular, a pointwise PID could yield valuable insights in contexts such as instance-level model interpretability, attribution analysis, or context-sensitive decision-making, where global averages may fail to capture the heterogeneity of information contributions across samples. However, the development of a sound theoretical framework for such a decomposition remains an open research problem, requiring new mathematical tools and likely new conceptual foundations. To the best of our knowledge, a fully general pointwise formulation of the PID components—particularly one that adheres to the axiomatic foundations of the framework—has not been rigorously established in the literature.
> > > Accordingly, while we recognize and appreciate the importance of this perspective, we believe that a rigorous treatment of pointwise or groupwise PID components falls outside the scope of the present work. We thus leave this as an important and compelling direction for future research.
> > >
> > > [1] Williams et al. “Nonnegative decomposition of multivariate information” (2010).

---

### Official Review · Reviewer_wgD1 · 2025-03-06

**Overall Recommendation:** 4

**Summary:**

This paper presents an empirical investigation of alignment between models with possibly different architectures and trained over different modalities. The authors investigate under which conditions the so-called Platonic Representation Hypothesis is likely to arise based on the heterogeneity of the data modalities and uniqueness in information. The empirical findings over multiple simulated data and a multi-modal benchmark reveal that alignment is not necessarily correlated to an increase in model performance, hence establishing that alignment between models can arise only under specific experimental conditions.

**Claims And Evidence:**

The main claim is that the two axes the authors have proposed to measure, namely uniqueness of information and heterogeneity, are responsible for more or less alignment in trained models. This is an interesting proposal that relates to other studies in catastrophic forgetting in continual learning, see e.g. [1]. Authors provide evidence of this relation in several synthetic datasets, generated according to these two axes of variation, and on real-world data and baselines.
The main investigation is to uncover if alignment correlates with model performance and scale. Overall, the evidence supports the claims that this depends on uniqueness and the modality gap.

It remains open whether and how models trained on larger datasets (consisting of multiple degrees of uniqueness) can express a higher degree of alignment. This is a more challenging scenario to test, worth spelling out in the conclusions.

**Essential References Not Discussed:**

N/A

**Experimental Designs Or Analyses:**

I focused more on synthetic experiments to understand the core message there. Why do the authors choose a non-linear transformation only for the second modality? Would it have been sensible to have it also for the first modality?

I suggest including random baselines when alignment is measured and plotted (RQ1).

**Methods And Evaluation Criteria:**

The methods and evaluation are clear for the synthetic experiments.
I struggled a bit to understand how authors investigate the real-world datasets and models. I require further clarifications from the authors that can help in reading and assessing the quality of their evaluation, see questions.
Overall, I'm leaning positively towards the analysis the authors conducted.

**Other Comments Or Suggestions:**

One minor note: I do not entirely understand how the bottom part with triangles and circles should be interpreted.

There is a repetition at the beginning of section 3, from line 118 onwards. The same sentence appears in line 65.

**Other Strengths And Weaknesses:**

Figures are helpful to understand the message.

**Questions For Authors:**

About real-world experiments:
1) How do you measure uniqueness for MOSEI, MOSI, URFUNNY, etc? Not entirely clear how this is evaluated from human annotation
2) How is it the case that perturbations of the input correspond to changing uniqueness? This aspect is not clear and is a bit toy. There is the risk that perturbed strings and images can create out-of-distribution inputs for both vision and language models. Is it sensible to expect there, where much information is distorted, to see any alignment at all?
3)  Lines 291-300 are not clear about the upper limit for alignment. How is this tested or referenced?

**Relation To Broader Scientific Literature:**

Understanding the datasets shared information or uniqueness is something relevant also in Continual Learning [1]. There, this information can lead to more or less catastrophic forgetting. Also, the multimodal setup where representations are compared resembles theoretical works on identifiability for the case of independent component analysis [2]. This connection can be helpful for new theory-oriented works.


[1] Toward Understanding Catastrophic Forgetting in Continual Learning, Nguyen et al. (2019)
[2] The Incomplete Rosetta Stone Problem: Identifiability Results for Multi-View Nonlinear ICA, Gresele et al. (2019)

**Theoretical Claims:**

N/A

---

> ### Author Rebuttal · Authors · 2025-04-01
>
> We thank the reviewer and are glad that they find our experimental evidence convincing. Below we address the reviewer’s comments and questions.
>
> **Under “Claims and Evidence:”**
>
> > It remains open whether and how models trained on larger datasets (consisting of multiple degrees of uniqueness) can express a higher degree of alignment.
>
> We present new results on MM-IMDb [1], a dataset for classifying movie genres with 25k paired images and texts. [Our results](https://tinyurl.com/5my72nnb) demonstrate that the relation between alignment and performance varies depending on the classification task (see response to B2rQ for more details), suggesting that the degree of alignment depends significantly on the downstream task.
>
> [1] Arevalo et al. “Gated multimodal units for information fusion” (2017).
>
>
> **Under “Experimental Designs Or Analyses:”**
>
> > I focused more on synthetic experiments to understand the core message there. … Would it have been sensible to have it also for the first modality?
>
> We acknowledge that there are many ways of defining heterogeneity, however, a benefit of leaving the first modality untransformed is that the representation that $E_1$ learns is an ideal one -- because it has a direct linear relationship with the labels -- aligning with this ideal representation could imply that the model has learned something really universal: a requirement for the Platonic hypothesis. Then if  $E_2$’s representation is highly aligned with $E_1$, we can infer that $E_2$ has learned to recover information that is comparable to the untransformed modality. Nevertheless, transforming both modalities may yield insightful results, and we leave the exploration of different types of heterogeneity to future work.
>
> > I suggest including random baselines when alignment is measured and plotted (RQ1).
>
> We have run experiments computing alignment between randomly initialized neural networks [here](https://tinyurl.com/5fwdecad). Results confirm that the alignment of these neural networks is constant with respect to uniqueness and that there is no correlation between alignment and performance  on average.
>
> **Under “Relation To Broader Scientific Literature:”**
>
> > Understanding the datasets shared information or uniqueness is something relevant also in Continual Learning [1].
>
> Thank you for bringing up these related works. We will include a discussion in our updated paper.
>
> **Under “Other Comments Or Suggestions:”**
> > One minor note: I do not entirely understand how the bottom part with triangles and circles should be interpreted.
>
> In Figures 1, the triangles and circles represent data from different modalities. In Figure 2, the triangles (and other shapes) also represent data from different modalities with varying degrees of heterogeneity. We will clarify this in the final version of our paper.
>
> > There is a repetition at the beginning of section 3, from line 118 onwards. The same sentence appears in line 65.
>
> Thank you for pointing this out. We will remove the redundancy.
>
> **Under “Questions For Authors:”**
> About real-world experiments:
>
> > How do you measure uniqueness for MOSEI, MOSI, URFUNNY, etc? Not entirely clear how this is evaluated from human annotation
>
> While we do not rely on exact estimates of uniqueness for the MultiBench datasets, past work [1] has sampled several data points and asked human annotators to rate the redundancy and uniqueness of each example. These ratings are shown to agree with computational estimates of redundancy and uniqueness for various MultiBench datasets.
>
> [1] Liang et al. “Quantifying & Modeling Multimodal Interactions: An Information Decomposition Framework” (2024)
>
> > How is it the case that perturbations of the input correspond to changing uniqueness? … Is it sensible to expect there, where much information is distorted, to see any alignment at all?
>
> We agree that our method of perturbing the Wikipedia caption dataset is not fully aligned with our definition of uniqueness. Hence, we provide new experiment results on MM-IMDb and improved our experiments on the Wikipedia-Image Text dataset. We use GPT-4 to synthesize text captions with unique information that is not present in the images, ensuring that the resulting datasets retain the semantics of real-world text and images. Our findings support our paper’s key claims (see response to obVJ for more details).
>
> > Lines 291-300 are not clear about the upper limit for alignment. How is this tested or referenced?
>
> While the theoretical upper bound for alignment, based on the HSIC metric, is 1, our empirical results (Figures 4 and 5, indicated by the red dot) show that the observed upper limit is significantly lower. We therefore conjecture that the maximum achievable alignment is constrained by the amount of shared information between the two modalities. We acknowledge that this argument is not rigorously formalized, and we will clarify this point in the updated version of the paper.

---

> > ### Comment · Reviewer_wgD1 · 2025-04-04
> >
> > Thank you for the extensive reply.
> >
> > >  Our results demonstrate that the relation between alignment and performance varies depending on the classification task (see response to B2rQ for more details), suggesting that the degree of alignment depends significantly on the downstream task.
> >
> > Can you elaborate on this? So, whether it is classification or image captioning or something else?
> > This does not answer the question of what happens if you have bigger and bigger datasets, because this is the standard case for training VLMs.
> >
> > > Nevertheless, transforming both modalities may yield insightful results, and we leave the exploration of different types of heterogeneity to future work.
> >
> > Yes, it would be useful to include that.
> >
> > > We have run experiments computing alignment between randomly initialized neural networks here. Results confirm that the alignment of these neural networks is constant with respect to uniqueness and that there is no correlation between alignment and performance on average.
> >
> > Thank you.
> >
> > > We agree that our method of perturbing the Wikipedia caption dataset is not fully aligned with our definition of uniqueness. Hence, we provide new experiment results on MM-IMDb and improved our experiments on the Wikipedia-Image Text dataset.
> >
> > This looks cool, thank you.

---

> > > ### Author Response · Authors · 2025-04-08
> > >
> > > We thank the reviewer for the suggestion of exploring different definitions of heterogeneity. We will include a discussion of how our analysis framework can be extended to different types of heterogeneity as future work.
> > >
> > > > Can you elaborate on this? So, whether it is classification or image captioning or something else? This does not answer the question of what happens if you have bigger and bigger datasets, because this is the standard case for training VLMs.
> > >
> > > By different downstream tasks, we meant that MM-IMDb has 23 categories of movies, and thus the multilabel classification task can be broken down into 23 binary classification tasks (e.g. classifying genre 1 vs. all other genres). We wanted to present a new use case of our analysis that would be relevant to larger datasets for which there are typically many downstream tasks (which can extend to generative tasks, such as image captioning as the reviewer pointed out). To clarify our answer to the original question of how alignment changes when there are potentially many degrees of uniqueness, we demonstrate that the alignment-performance correlation depends on the amount of unique information **that is task relevant**. In the case of MM-IMDb, even though the text modality can contain many degrees of uniqueness compared to the image (as the text summarizes the plot of the movie), not all of the additional information that the text provides about the plot would be useful to the given classification task. Therefore, our analysis would reveal for each task whether the degrees of uniqueness are task-relevant. Smaller linear fit slopes to alignment-performance scores suggest that aligning modalities is less helpful for certain tasks, in which case practitioners should focus on modeling unique information.

---

### Official Review · Reviewer_obVJ · 2025-03-12

**Overall Recommendation:** 2

**Summary:**

This paper mainly focus on analyzing the emergence of multimodal representation alignment. Alignment between cross-modal representations has been long regarded as an important factor of improving multimodal model performance. Some recent researches have found that independently trained unimodal models can be implicitly aligned. The authors aim at finding out the opportunity and reasons for alignment emergence, and whether such alignment is a indicator of performance. Through comprehensive synthetic and real-world dataset experiments, the author reach to several conclusions. 1. The alignment may not be universally beneficial. 2. Such alignment impacts on performance differently among datasets and tasks.

**Claims And Evidence:**

The authors mainly discuss the emergence of implicit alignment in multimodal training. There are basically following claims in this paper: 1. Under low uniqueness, the alignment is significantly correlated with performance and model capacity. However, when uniqueness increases, such relationship becomes much weaker. 2. Alignment alone is not a sufficient predictor of model performance especicially in multimodal settings of uniqueness and heterogeneity.

Although I have several concerns about the experimental designs and their support to the final conclusions, I generally agree with the claims within the paper. My major question concerns the heuristic of this paper on modern multimodal model design. Although the authors provide detailed experiments and analysis, the conclusions seem to be obvious and intuitive. The observation that alignment is less related to performance when uniqueness and heterogeneity increase is not much novel. In contrast, I am more curious about how such conclusions can impact on the design of modern models on various datasets or down-stream tasks, which however is less discussed thoughout the paper.

**Essential References Not Discussed:**

No other related works need to be mentioned.

**Experimental Designs Or Analyses:**

I appreciate the designs of the synthetic experiments. The uniqueness assessment and label generation are reasonable. However, the experimental setup for real benchmark seems to be unaligned with the problem settings. Accordingly, on the Wikipedia caption dataset, the uniqueness of text and image data is implemented by random deletion and Gaussian perturbation, which is actually injetced noise. Such design seems to be opposite to the definition of uniqueness in line 155-162 that "Uniqueness in modality quantifies the amount of information present in the first modality absent in the second but critical for the downstream task", since noise can not be crucial. Thus the conclusion from real-world experiments may not be convincing. I am also concerned about the setting of unsymmetrical encoders in line 190-195, that $E_1$ is simply a single-layer encoder while $E_2$ is a deep encoder of varying depth. While I've noticed the second modality is simulated by a nonlinear transformation, such design can lead to issue that $E_1$ can easily learn a good representation while the optimization of $E_2$ can be much harder.

**Methods And Evaluation Criteria:**

This paper is mainly analytical without giving a method. The evaluation metrics of the paper, for example, CKA and uniqueness is reasonable. However, in Fig. 4 and 5, the notation $r$ is never introduced before.

**Other Comments Or Suggestions:**

No other suggestions.

**Other Strengths And Weaknesses:**

While the paper is written straight-forward, here are several weaknesses in the writing that should be improved. For instance, the introduction of CKA is overlength in the second page. Since this is a contribution of previous works, details of this part is recommended to be placed in the Appendix. The explanation of $x_r$ and $x_u$ is duplicate in line 207-209 with line 171-176. The bijection $\phi$ in line 212 has also be introduced before.

**Questions For Authors:**

Please refer to the former parts. My major concerns include the novelty of the conclusions on implicit alignment, the impact of such conclusions on modern multimodal model designs and experimental designs.

**Relation To Broader Scientific Literature:**

Please refer to the former parts of the review, the impact of alignment conclusions on the design of modern models on various datasets or down-stream task is less discussed in this paper.

**Theoretical Claims:**

This paper is mainly analytical without giving theoretical claims.

---

> ### Author Rebuttal · Authors · 2025-04-01
>
> We thank the reviewer for the review. Below we address their questions and concerns.
>
> **Under “Claims and Evidence”:**
> >  My major question concerns the heuristic of this paper on modern multimodal model design … which however is less discussed thoughout the paper.
>
> To the best of our knowledge, our analysis of the emergence of alignment across the dimensions of uniqueness and heterogeneity is novel and fills an important gap in the literature on cross-modal alignment. While prior work—such as the Platonic Representation Hypothesis [1]—suggests that alignment tends to emerge with increasing data scale and serves as an indicator of good performance, these claims have not been rigorously examined across key characteristics of multimodal data. In this paper, we critically evaluate these assumptions and argue that while alignment may indeed correlate with performance in settings where modalities share high redundancy (i.e., low uniqueness), this relationship breaks down when the modalities are more distinct. In such scenarios, increased alignment does not necessarily translate to better downstream performance. We believe this insight is not only novel but also practically useful, as it encourages practitioners to reconsider alignment strategies in cases where they may be counterproductive.
>
> We additionally explore the application of alignment-performance correlation for quantifying the information content of downstream tasks. Specifically, we present results on MM-IMDb [1], a dataset for classifying movie genres with image and text modalities. [Our results](https://tinyurl.com/5my72nnb) demonstrate that the relation between alignment and performance varies depending on the classification task (see response to B2rQ for more details), which can inform practitioners when aligning modalities is beneficial.
> [1] Arevalo et al. “Gated multimodal units for information fusion” (2017).
>
> **Under Methods And Evaluation Criteria:**
> > This paper is mainly analytical without giving a method. … However, in Fig. 4 and 5, the notation  is never introduced before.
>
> Thank you for the feedback. We will update our paper to define Alignment as unbiased CKA and Unique as the number of unique features used in computing the label.
>
> **Under Experimental Designs Or Analyses:**
>
> > I appreciate the designs of the synthetic experiments. The uniqueness assessment and label generation are reasonable. However, the experimental setup for real benchmark seems to be unaligned with the problem settings. … the conclusion from real-world experiments may not be convincing.
>
> We agree that our method of perturbing the Wikipedia caption dataset is not fully aligned with our definition of uniqueness. To ensure that the perturbed dataset retains the semantics of real-world text and images, we provide new experiment results that leverage GPT-4 to synthesize text captions with unique information that is not present in the images. We keep the original image data without any additional noise. We upload our perturbed text data and code for generating the perturbations [here](https://tinyurl.com/8vxt9hby). For each (image, text) pair in the original dataset, we prompt GPT-4 to produce 10 captions with increasing levels of uniqueness: 10%, 20%, … 100%, such that the final caption contains only information that is unique to the text. As uniqueness is already introduced in the text, we keep the original images in the Wikipedia caption dataset. Using a pretrained sentence BERT model to quantify semantic similarity between the original caption and the GPT-4 captions, we find that the average semantic similarity monotonically decreases as the level of uniqueness increases. We compute the alignment between various types of vision models and LLMs. Our updated results support both claims: 1) The maximum alignment decrease with increased uniqueness. [see figure here](https://tinyurl.com/yeezwdxu) and 2) The slope of the fitted line to the alignment and performance scores decreases with increased uniqueness, showing that the relation between alignment and performance weakens. [see figure here](https://tinyurl.com/4ynj2td3).
>
> > I am also concerned about the setting of unsymmetrical encoders in line 190-195 … while the optimization of $E_2$ can be much harder.
>
> We provide additional experiment results demonstrating that our results are unchanged when $E_1$ is a higher depth. [Here](https://tinyurl.com/4uwe4zkm), we change the depth of $E_1$ to 2 and 3 and find that the results are not significantly changed. We hypothesize that because $E_1$ is trained on the untransformed modality, $E_1$ will remain relatively easy to optimize even as the depth increases. We will include these results in our updated paper.
>
> **Under Other Strengths And Weaknesses:**
>
> > While the paper is written straight-forward, here are several weaknesses in the writing that should be improved.
>
> Thank you for the feedback. We will revise our paper accordingly.

---

> > ### Comment · Reviewer_obVJ · 2025-04-02
> >
> > Thank you for your explanation and detailed experiments. My concerns about unsymmetrical encoder depth has been solved. The misalignment between settings and experiments has also been made up through additional experiments. Both experiments are expected to be added during further revision.
> >
> > On the other side, my concerns about the novelty and practicability of the alignment analysis in this paper is still under underexplored. I am mostly convinced about the explored relationship between alignment with uniqueness and heterogeneity, as stated by in both the main paper and rebuttal. However, practitioners are more concerned about the impact of such relation on pratical usage. For instance, as facing a real-world large scale scenario, when and how should we measure such relation, how should we adjust the training procedure according to the relation. These questions are less discussed during the paper and rebuttal. Reviewer B2rQ seems to share similar concerns that "No new method is proposed, which limits the contribution of this paper."
> >
> > In conclusion, I will raise my score to 2 for detailed experiments, and will carefully consider my score if the authors can make further explanation or there is any comments made by other reviewers.

---

> > > ### Author Response · Authors · 2025-04-08
> > >
> > > We appreciate the reviewer's response and the opportunity to further clarify our work. Below, we address their concern regarding the practicality of our analysis by providing additional experimental results. While we recognize the importance of practical implications, we would like to respectfully emphasize that the primary contribution of our study lies in the systematic refutation of the PRH—an aspect that, to the best of our knowledge, has not been previously established. Although our conclusion may align with intuitive expectations, we believe that this does not diminish the novelty of formally demonstrating that the PRH does not universally hold.
> > >
> > > > Thank you for your explanation … Both experiments are expected to be added during further revision.
> > >
> > > We will add the experiments to our updated paper.
> > >
> > > > On the other side, my concerns about the novelty and practicability of the alignment analysis in this paper is still under underexplored. ... For instance, as facing a real-world large scale scenario, when and how should we measure such relation, how should we adjust the training procedure according to the relation. These questions are less discussed during the paper and rebuttal. Reviewer B2rQ seems to share similar concerns that "No new method is proposed, which limits the contribution of this paper."
> > >
> > > We present the following use case for our analysis. Consider a practical setting where there is a large dataset of paired input data, but only a small subset of the dataset has labels for downstream tasks, due to the cost of annotation. An important problem is how can a practitioner utilize the supervision from the data subset while still ensuring good generalization by leveraging the unlabeled paired data? One approach is to finetune a pretrained model using both supervised loss and an explicit alignment objective, such as the CLIP loss. However, an important question comes up: how should the contribution of the supervised and alignment losses be balanced to maximize performance? The loss takes the form of  $\mathcal{L} = \mathcal L_{sup} + w * \mathcal L_{\text{CLIP}} $ From our analysis, we know that the “ideal” amount of alignment is dataset and task-specific. Specifically, alignment-performance correlations have a direct algorithmic implication: if the alignment-performance correlation is small, then performance degrades or does not change when increasing the weight on the explicit alignment objective. Conversely, when the alignment-performance correlation is larger, performance should increase with larger weight on the alignment objective.
> > >
> > > To test this idea, we run experiments on the MM-IMDb dataset on 10 different binary classification tasks, where we sample 1024 labeled examples for each of the train, validation and test sets to simulate the data scarce scenario (in comparison to the original dataset size of 25k examples). The alignment-performance correlations can be easily computed with pretrained vision and language models using the sampled data. We start with vision and language encoders pretrained with CLIP and finetune the models with $\mathcal L$, where the weight on the alignment objective varies in $w \in \{0, 0.1, 0.25, 0.5, 1.0, 2.0, 5.0, 10.0, 50.0, 100.0\}$. In agreement with our analysis, [our results](https://tinyurl.com/mwpresr2) demonstrate that on the categories with lower alignment-performance correlation, increasing $w$ leads to worse performance, whereas for classes with higher-performance correlations, high values of $w$ improve performance. These results show that quantifying the relation between alignment-performance, **even with unimodal models that are not explicitly aligned**, is useful for practitioners when deciding how much to explicitly align the modalities. We envision that future work would make use of alignment-performance correlations to automatically determine weight on the alignment loss for each downstream task, making it possible to train on many tasks simultaneously without a combinatorially expensive hyperparameter search (if there are 23 tasks and 8 discrete values of $w$, there are 8$^{23}$  combinations of parameters to search over).
> > >
> > > We note that while we experiment with CLIP, our proposed framework is agnostic to the specific alignment loss. This is because our contribution is the **balance between a supervised objective that directly optimizes some downstream performance and an alignment metric, which is interchangeable.** Therefore, alignment-performance correlations remain useful regardless of whether the modalities are aligned through CLIP or a different approach such as FactorCL [1], as brought up by reviewer B2rQ.
> > >
> > > [1]  Liang et al. “Factorized contrastive learning: Going beyond multi-view redundancy” (2023).

---

### Official Review · Reviewer_B2rQ · 2025-03-14

**Overall Recommendation:** 3

**Summary:**

This paper empirically investigates when and why implicit alignment emerges, and whether alignment consistently predicts task performance, finding that both depend critically on modality similarity and the redundancy or uniqueness of the information provided.

**Claims And Evidence:**

The analysis that is conducted highly depends on the alignment quantification.
Is the used metric, HSIC, sufficient to reflect the alignment quality?  Such kernel-based metric is highly sensitive to the chosen kernel, sample size and other hyper parameters. If the metric cannot truly reflect the alignment level, the experiments, like Emergence of alignment across heterogeneity and uniqueness, are questionable.

**Essential References Not Discussed:**

Most relevant papers are discussed.

**Experimental Designs Or Analyses:**

The experimental analysis of the synthetic dataset is interesting. However, more comprehensive experimental analysis should be performed on a large scale real-world dataset rather than a subset of MultiBench to study Emerge property.

**Methods And Evaluation Criteria:**

No new method is proposed.

**Other Comments Or Suggestions:**

see my weakness.

**Other Strengths And Weaknesses:**

The motivation for analyzing the alignment is interesting.

Weakness:
1. Is the used metric, HSIC, sufficient to reflect the alignment quality?  Such kernel-based metric is highly sensitive to the chosen kernel, sample size and other hyper parameters. If the metric cannot truly reflect the alignment level, the experiments, like Emergence of alignment across heterogeneity and uniqueness, are questionable.
2. No new method is proposed, which limits the contribution of this paper.
3. Most experimental analysis is based on the synthetic datasets, which is not convincing. More comprehensive experimental analysis should be performed on large scale real-world dataset, rather than a subset of MultiBench to study Emerge property.
4. It would be interesting to see the quantification of the uniqueness level in the real-world dataset.
5. [1] proposes that different random initializations could also cause a modality gap. Will this affect the conclusion of this paper?


[1]. Liang, Victor Weixin, et al. "Mind the gap: Understanding the modality gap in multi-modal contrastive representation learning." Advances in Neural Information Processing Systems 35 (2022): 17612-17625.

**Questions For Authors:**

see my weakness.

**Relation To Broader Scientific Literature:**

The results are relevant to the analysis of multimodal alignment.

**Theoretical Claims:**

No theoretical claim.

---

> ### Author Rebuttal · Authors · 2025-04-01
>
> We thank the reviewer for the constructive criticism and are glad that they find our analysis interesting. Below we address the reviewer’s questions and concerns.
>
> **Under “Claims And Evidence”:**
>
> > Is the used metric, HSIC, sufficient to reflect the alignment quality? … is highly sensitive to the chosen kernel, sample size and other hyper parameters.
>
> We believe that the HSIC metric is sufficient to capture alignment quality, as we use a specific linear kernel consistently across all experiments. Moreover, this kernel has no hyperparameters, making it the simplest choice. To verify the robustness of our results, we also evaluate them using alternative alignment metrics. We perform additional experiments on the synthetic data with additional alignment metrics and with different sample sizes which demonstrate that our findings are robust to hyperparameters and consistent across different metrics. Specifically, we report results with unbiased CKA with a linear kernel (our original alignment metric), unbiased CKA with a RBF kernel, Mutual KNN [2], SVCCA [3] and run all metrics with 256, 512 (our original sample size), 1024 data points. We report our [updated results here](https://tinyurl.com/yfzwzs2s). For all metrics and batch sizes, the maximum alignment decreases with increasing uniqueness as well as increasing heterogeneity. Additionally,  the relation between alignment, performance, and depth are consistent across different batch sizes. Across all alignment metrics, performance and depth are positively correlated over different uniqueness values, whereas alignment and performance as well as alignment and depth correlations can be weak or negative for increased uniqueness, indicating that our findings are robust to different kernels and alignment metrics.
>
> [1] Kornblith et al. “Similarity of Neural Network Representations Revisited” (2019).
> [2] Huh et al. "The platonic representation hypothesis" (2024).
> [3] Raghu et al. “SVCCA: Singular Vector Canonical Correlation Analysis for Deep Learning Dynamics and Interpretability” (2017).
>
> **Under “Other Strengths and Weaknesses”:**
>
> > No new method is proposed, which limits the contribution of this paper.
>
> While our paper does not propose a new method, we believe that our contributions are significant -- see our response to reviewer obVJ for a more in-depth discussion.
>
> > Most experimental analysis is based on the synthetic datasets … rather than a subset of MultiBench to study Emerge property.
>
> We would like to emphasize that MultiBench datasets (used in Section 6) are real-world, with CMU-MOSEI and UR-FUNNY containing 22k and 16k video snippets respectively. In addition, we provide new experiment results on MM-IMDb and improved our experiments on the Wikipedia-Image Text dataset. We use GPT-4 to synthesize text captions with unique information that is not present in the images, ensuring that the resulting datasets retain the semantics of real-world text and images. Our findings support our paper’s key claims (see response to obVJ for more details).
>
> > It would be interesting to see the quantification of the uniqueness level in the real-world dataset.
>
> We agree that quantifying uniqueness is an interesting direction, and our results have shown the potential for alignment-performance correlation to be used for quantification. While different pairs of modalities have varying levels of heterogeneity, which can make it difficult to quantify uniqueness across datasets, we propose that alignment-performance correlations can quantify information content between different downstream tasks within a given multimodal dataset. We present new results on MM-IMDb [1], a dataset for classifying movie genres with image and text modalities. Each movie can be labeled with 1 or more genres, and there are 23 classes. We compute cross-modal alignment using various vision models and language models. To measure performance, we train linear layers on the last layer hidden representations of the language models, resulting in F1-scores for each class. [Our results](https://tinyurl.com/5my72nnb) demonstrate that the relation between alignment and performance varies depending on the classification task — we see that the slope of the linear fit to alignment and performance scores is weak or even negative, suggesting that for certain movie genres, there is greater task relevant information that is unique to the language modality.
>
> [1] Arevalo et al. “Gated multimodal units for information fusion” (2017).
>
> > [1] proposes that different random initializations could also cause a modality gap. Will this affect the conclusion of this paper?
>
> We would like to clarify that our experiments on synthetic data are run with 5 different seeds, and for our experiments on MultiBench datasets, we compute alignment-performance correlation over 3 seeds. Hence, we believe that our results are robust to initializations.

---

> > ### Comment · Reviewer_B2rQ · 2025-04-07
> >
> > Thanks for the detailed reply.
> >
> > I have a similar concern with the reviewer obVJ about "the impact of such relation on practical usage". I agree that the contribution of theoretical analysis could be significant. However, a more theoretical analysis and demonstration of practical usage are necessary to make the paper sound.
> >
> > The effect of uniqueness and shared information among different modalities has been quantified by FactCL [1]. I apologize for not bringing this paper in the first place. FactCL uses mutual information for analysis and measuring the impact of uniqueness in different modalities. Their quantification of "uniqueness" leads to a new novel method for alignment and demonstrates significantly better performance on both synthetic and real-world MultiBench datasets. How to utilize the proposed correlation relationships for practical usage is a big concern.
> >
> > Moreover, I am not sure if the correlation analysis is sufficient as correlation is not causality. A strong correlation does not tell you whether one variable causes changes in the other or whether both are driven by some unobserved factor.
> >
> >
> > I will carefully consider my rating and appreciate if the authors could make further clarifications.
> >
> >
> > [1]. Liang P P, Deng Z, Ma M Q, et al. Factorized contrastive learning: Going beyond multi-view redundancy[J]. Advances in Neural Information Processing Systems, 2023, 36: 32971-32998.

---

> > > ### Author Response · Authors · 2025-04-08
> > >
> > > We appreciate the reviewer's response and the opportunity to further clarify our work. Below, we address concerns on the practicality of our analysis with additional experimental results. While we recognize the importance of practical implications, we would like to respectfully emphasize that the primary contribution of our study lies in the systematic refutation of the PRH—an aspect that, to the best of our knowledge, has not been previously established. Although our conclusion may align with intuitive expectations, we believe that this does not diminish the novelty of formally demonstrating that the PRH does not universally hold.
> > >
> > > > I have a similar concern with the reviewer obVJ about "the impact of such relation on practical usage" … necessary to make the paper sound.
> > >
> > > To demonstrate the practical usage of our analysis, we present the following use case. Consider a setting where there is a large dataset of paired input data, but only a small subset of the dataset has labels for downstream tasks, due to the cost of annotation. An important problem is how can a practitioner utilize the supervision from the data subset while still ensuring good generalization by leveraging the unlabeled paired data? One approach is to finetune a pretrained model using both supervised loss and an explicit alignment objective, such as the CLIP loss. However, an important question comes up: how should the contribution of the supervised and alignment losses be balanced to maximize performance? The loss takes the form of  $\mathcal{L} = \mathcal L_{sup} + w * \mathcal L_{\text{CLIP}} $ From our analysis, we know that the “ideal” amount of alignment is dataset and task-specific. Specifically, alignment-performance correlations have a direct algorithmic implication: if the alignment-performance correlation is small, then performance degrades or does not change when increasing the weight on the explicit alignment objective. Conversely, when the alignment-performance correlation is larger, performance should increase with larger weight on the alignment objective.
> > >
> > > To test this idea, we run experiments on the MM-IMDb dataset on 10 different binary classification tasks, where we sample 1024 labeled examples for each of the train, validation and test sets to simulate the data scarce scenario (in comparison to the original dataset size of 25k examples). The alignment-performance correlations can be easily computed with pretrained vision and language models using the sampled data. We start with vision and language encoders pretrained with CLIP and finetune the models with $\mathcal L$, where the weight on the alignment objective varies in $w \in \{0, 0.1, 0.25, 0.5, 1.0, 2.0, 5.0, 10.0, 50.0, 100.0\}$. In agreement with our analysis, [our results](https://tinyurl.com/mwpresr2) demonstrate that on the categories with lower alignment-performance correlation, increasing $w$ leads to worse performance, whereas for classes with higher-performance correlations, high values of $w$ improve performance. These results show that quantifying the relation between alignment-performance, **even with unimodal models that are not explicitly aligned**, is useful for practitioners when deciding how much to explicitly align the modalities. We envision that future work would make use of alignment-performance correlations to automatically determine weight on the alignment loss for each downstream task, making it possible to train on many tasks simultaneously without a combinatorially expensive hyperparameter search (if there are 23 tasks and 8 discrete values of $w$, there are 8$^{23}$ combinations of parameters to search over).
> > >
> > > > The effect of uniqueness and shared information … FactCL uses mutual information for analysis and measuring the impact of uniqueness in different modalities.
> > >
> > > As discussed in our above response, our analysis is useful for understanding how representation alignment relates to performance on some downstream task, and therefore, practitioners would use our analysis to design a better training objective that optimally balances explicit alignment with direct optimization of downstream performance. While we experiment with CLIP, our proposed framework is agnostic to the specific alignment loss. We believe our work is complementary to the literature on improving explicit alignment objectives for paired, unlabeled data. We will clarify this difference in our updated paper.
> > >
> > > > Moreover, I am not sure … by some unobserved factor.
> > >
> > > We agree that correlation is not causality. However, we indeed show that the alignment-performance correlations have direct implications on how practitioners should balance explicit alignment with supervised learning. In addition, we have extensive experiments on synthetic and real-world settings, showing that factors such as uniqueness and heterogeneity impact the relation between alignment and performance.
> > >
> > > [1]  Liang et al. “Factorized contrastive learning: Going beyond multi-view redundancy” (2023).

---

### Decision · Program_Chairs · 2025-05-01

**Decision:**

Accept (poster)

**Comment:**

Four reviewers submitted reviews for this submission. In the pre-rebuttal phase, reviewers raised some concerns such as:

- no new method is proposed
- is the HSIC metric used sufficient for reporting numbers
- most results are on synthetic datasets
- impact of relations and practicality of the method
- some questions to improve clarity

All reviewers acknowledged the rebuttal and after the post-rebuttal phase, the final ratings are two accepts, a weak accept and a weak reject. Reviewers acknowledged that most of their concerns have been resolved after the post-rebuttal phase. Although a point of practical usability remained, during discussion with reviewers, it was identified that the paper contains sufficient interesting insights and advances our scientific understanding significantly and therefore this practicability aspect can be left for future work. Therefore, the decision is to recommend the acceptance of the paper. Authors are recommended to incorporate important reviewer's comments the final version.